# An essential role for tungsten in the ecology and evolution of a previously uncultivated lineage of anaerobic, thermophilic Archaea

Steffen Buessecker [1,14✉], Marike Palmer [2,14✉], Dengxun Lai [2], Joshua Dimapilis[3], Xavier Mayali [4], Damon Mosier [3,5], Jian-Yu Jiao [6], Daniel R. Colman[7], Lisa M. Keller[7], Emily St. John [8], Michelle Miranda[3], Cristina Gonzalez[3], Lizett Gonzalez[3], Christian Sam[3], Christopher Villa[3], Madeline Zhuo[3], Nicholas Bodman[3], Fernando Robles[3], Eric S. Boyd[7], Alysia D. Cox [9], Brian St. Clair[9], Zheng-Shuang Hua [10], Wen-Jun Li [6,11], Anna-Louise Reysenbach[8], Matthew B. Stott [12], Peter K. Weber [4], Jennifer Pett-Ridge [4,13], Anne E. Dekas [1], Brian P. Hedlund [2] & Jeremy A. Dodsworth[3✉]

Trace metals have been an important ingredient for life throughout Earth's history. Here, we describe the genome-guided cultivation of a member of the elusive archaeal lineage *Caldarchaeales* (syn. *Aigarchaeota*), *Wolframiiraptor gerlachensis*, and its growth dependence on tungsten. A metagenome-assembled genome (MAG) of *W. gerlachensis* encodes putative tungsten membrane transport systems, as well as pathways for anaerobic oxidation of sugars probably mediated by tungsten-dependent ferredoxin oxidoreductases that are expressed during growth. Catalyzed reporter deposition-fluorescence in-situ hybridization (CARD-FISH) and nanoscale secondary ion mass spectrometry (nanoSIMS) show that *W. gerlachensis* preferentially assimilates xylose. Phylogenetic analyses of 78 high-quality *Wolframiiraptoraceae* MAGs from terrestrial and marine hydrothermal systems suggest that tungsten-associated enzymes were present in the last common ancestor of extant *Wolframiiraptoraceae*. Our observations imply a crucial role for tungsten-dependent metabolism in the origin and evolution of this lineage, and hint at a relic metabolic dependence on this trace metal in early anaerobic thermophiles.

---

[1] Department of Earth System Science, Stanford University, Stanford, CA, USA. [2] School of Life Sciences, University of Nevada, Las Vegas, Las Vegas, NV, USA. [3] Department of Biology, California State University, San Bernardino, CA, USA. [4] Physical and Life Sciences Directorate, Lawrence Livermore National Laboratory, Livermore, CA, USA. [5] Department of Geoscience, University of Calgary, Calgary, AB, Canada. [6] State Key Laboratory of Biocontrol, Guangdong Provincial Key Laboratory of Plant Resources and Southern Marine Science and Engineering Guangdong Laboratory (Zhuhai), School of Life Sciences, Sun Yat-Sen University, Guangzhou, PR China. [7] Department of Microbiology and Cell Biology, Montana State University, Bozeman, MT, USA. [8] Department of Biology, Portland State University, Portland, OR, USA. [9] Department of Chemistry and Geochemistry, Montana Technological University, Butte, MT, USA. [10] Department of Environmental Science and Engineering, University of Science and Technology of China, Hefei, PR China. [11] College of Fisheries, Henan Normal University, Xinxiang, PR China. [12] School of Biological Sciences, University of Canterbury, Christchurch, New Zealand. [13] Life & Environmental Sciences Department, University of Merced, Merced, CA, USA. [14] These authors contributed equally: Steffen Buessecker, Marike Palmer. ✉email: sbuessecker@stanford.edu; marike.palmer@unlv.edu; JDodsworth@csusb.edu

Evidence is mounting that some of the most ancient life forms inhabited terrestrial hot springs present on the anoxic early Earth[1–4]. Such habitats could have mobilized and concentrated nutrients including trace metals essential to thermophiles, but that differ from the metals essential to extant aerobic mesophiles[4]. Contemporary terrestrial hot springs are reasonable analogs for those on early Earth, often exhibiting low reduction potentials and sulfur-rich anoxic conditions[5]. In the presence of sulfide, several trace metals, including molybdenum (Mo), copper (Cu), nickel (Ni), and cobalt (Co) react with sulfide and precipitate as solid phases[6], rendering them inaccessible to microbial cells. This, in turn, could have represented a strong selective pressure to evolve metalloenzymes that were dependent on less thiophilic trace metals, such as manganese (Mn) and tungsten (W).

In modern hot springs, most microbial lineages lack a cultured representative[7,8], making the assessment of nutrient requirements difficult. Although cultivation-independent "omics" techniques hold tremendous promise in assessing the functional potential and activity of communities in their natural state, results from such studies are limited by what is known of characterized proteins from cultured organisms. Yet, for uncultivated lineages, many of the proteins are poorly annotated, often leading to an incomplete picture of their ecology and physiology. As such, there is a great need to bring these organisms into the culture so that the traits that permit the habitation of terrestrial hot springs, both today and in the geological past, can be examined.

*Candidatus* Caldarchaeales (syn. Aigarchaeota) is a yet-uncultivated lineage detected in high-temperature environments that was first described based on a genomic assembly from a thermal stream in a subsurface gold mine in Japan[9,10]. Although originally designated as candidate phylum Aigarchaeota, this lineage has been reassigned to the order *Ca.* Caldarchaeales within the phylum *Thermoproteota* and class *Nitrososphaeria*[11]. For simplicity, we use this latter taxonomy without the *Candidatus* designation for uncultivated taxa. *Caldarchaeales* are commonly found in hydrothermal and geothermal ecosystems[8], where they can be abundant[12–14]. Metagenomics and 16S rRNA gene amplicon surveys have previously shown a high abundance of *Caldarchaeales* in sediments of Great Boiling Spring (GBS, Nevada, USA)[15]. In particular, members of the putative genus "Aigarchaeota Group 4" (AigG4)[8] were abundant in in-situ GBS enrichments cultivated on lignocellulosic substrates such as corn stover[16,17]. AigG4 species from GBS and from hot springs in Tengchong, China (GMQ_bin_10 and JZ_bin_10 in ref. [12]) have been predicted to be anaerobic heterotrophs that degrade polysaccharides and peptides[13,14]. However, these predictions have yet to be experimentally validated and other physiological traits of AigG4 remain unknown.

Here, we report the enrichment and stable laboratory growth of an AigG4 species from GBS we designate *Wolframiiraptor gerlachensis*. After initial unsuccessful attempts to grow the species without amending growth media with spring water, analysis of a metagenome-assembled genome (MAG) of *W. gerlachensis* suggested possible dependence on the biologically rare trace metal W, as evidenced by six annotated W-dependent ferredoxin oxidoreductases that could play central roles in anaerobic carbohydrate degradation. Indeed, stable cultivation of *W. gerlachensis* in a synthetic medium containing corn stover or a monosaccharide mix required the addition of W. NanoSIMS analysis demonstrated that xylose is the carbohydrate monomer preferred by *W. gerlachensis* in in-situ corn stover-degrading enrichments. Based on these experimental results and the genomic potential of the MAG, we propose that W is essential for carbohydrate metabolism and plays a critical role in one or more of the annotated W-dependent ferredoxin oxidoreductases, which could replenish

reducing equivalents for a type-4 [NiFe] membrane-bound hydrogenase, producing a proton motive force. Using environmental metagenomes from three continents and a marine hydrothermal vent, we present evidence that both putative tungstate ($WO_4^{2-}$) transporters and tungstoenzymes were ancestral in this lineage, herein designated the family *Wolframiiraptoraceae*, and trace their evolution among 78 high-quality MAGs representing four genera and 11 species. While it is known that W is important for several archaeal groups[18–21], including methanogens, representatives of the *Thermococcales* and *Thermoproteales* are the only known archaea that require W for growth[18]. Our investigation is unique in showing long-term W-dependent growth dynamics in a mixed culture, identifying W-requiring archaea outside of the *Methanobacteriota* (formerly Euryarchaeota), and demonstrating large-scale expansions of W-dependent enzymes in a microbial lineage.

## Results

**A stable enrichment culture of an anaerobic member of the *Caldarchaeales*.** Because *W. gerlachensis* was previously enriched on corn stover incubated in GBS[17], we deployed new corn stover in-situ enrichments and incubated them for 6 months (Supplementary Fig. 1). The water above the incubation site was 86.5 °C, with a pH of 7.31 at the time the enrichment was deployed. After incubation, *W. gerlachensis* was present in the in-situ enrichment at ~3% of the total community based on 16S rRNA gene tag analysis (Fig. 1a), similar to previous results[17], along with other taxa previously observed in enrichments or GBS sediments[16,17]. Based on the reproducible enrichment of *W. gerlachensis* on corn stover, and the genome-predicted anaerobic lifestyle[13,14], we used the in-situ enrichment to inoculate 13 different anaerobic media that contained corn stover and/or xyloglucan and keratin as carbon sources. Some enrichments also contained previously sampled GBS spring water that was filtered (0.2 μm) and autoclaved at a 1:1 ratio with synthetic medium (Supplementary Table 1) to test if unknown components of the spring water were required for growth. The abundance of *W. gerlachensis* under culture conditions without added spring water dropped below the level of detection. However, *W. gerlachensis* was maintained after multiple transfers in two cultures with added spring water (A8 and A10) at concentrations of ~0.25–1.0 × 10^6 16S rRNA gene copies mL^−1 culture, with a relative abundance of ~0.5–7% (Fig. 1b).

The two culture conditions that maintained stable populations of *W. gerlachensis* contained GBS spring water but were otherwise identical to synthetic media enrichments where *W. gerlachensis* was not maintained (compare A10 with A1, and A8 with A3; Supplementary Table 1). Additional experiments determined that nitrate (0.1 mM) was inhibitory to *W. gerlachensis* (explaining the lack of growth in culture A9) and that robust growth was dependent on anoxic conditions and on the presence of corn stover rather than other substrates (Supplementary Figs. 1 and 2 and Supplementary Table 1). As a result, most subsequent experiments were performed using a variation of the A10 medium (A10X2) with corn stover as the main carbon source under anoxia. Under these conditions, *W. gerlachensis* has been maintained at an abundance of 10^6–10^7 16S rRNA gene copies mL^−1 culture for over 4 years. To our knowledge, these represent only the second reported laboratory cultures containing members of the *Caldarchaeales*[22].

**W is essential to maintain the growth of *W. gerlachensis*.** Since GBS spring water was necessary for the maintenance of *W. gerlachensis* cultures (Fig. 1b and Supplementary Table 1), we compared the composition of the synthetic medium to GBS spring water using

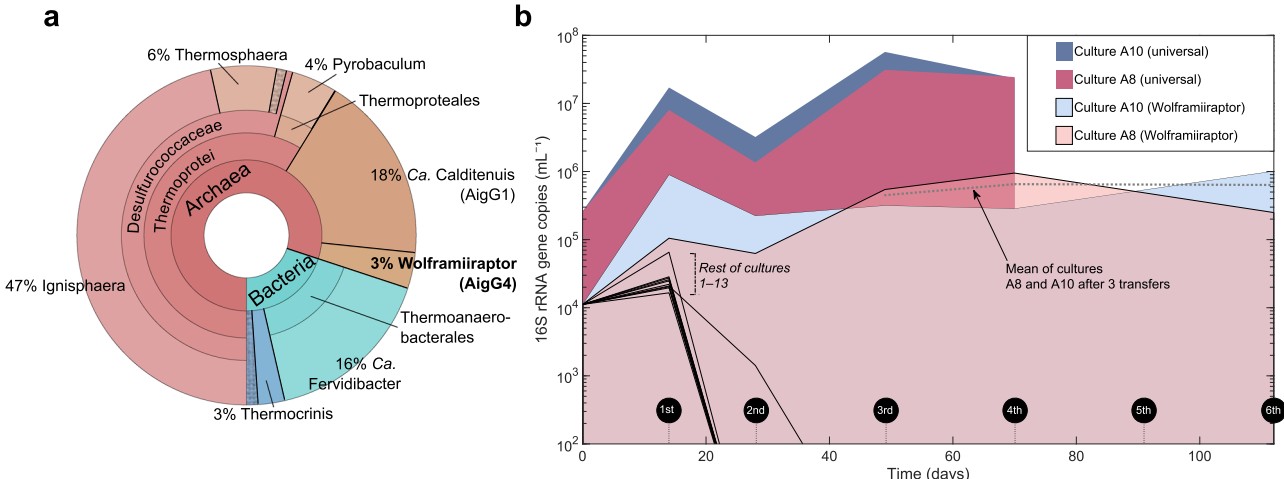

**Fig. 1 16S rRNA gene-based composition of *Caldarchaeales*-containing cultures. a** Taxonomic composition of a sediment in-situ enrichment from Great Boiling Spring inferred from 16S rRNA gene tag sequencing. **b** *Wolframiiraptor* (*W. gerlachensis*) and total microbial abundances within enrichment cultures, as determined by qPCR. The absolute abundance and proportion of *W. gerlachensis* was stable after three transfers. Curves marked by dark coloration represent 16S rRNA gene quantities of the total community and curves marked by light coloration show 16S rRNA gene quantities belonging to *W. gerlachensis* in cultures A8 and A10 (all other cultures have no colored fields). Source data are provided as a Source Data file.

inductively coupled plasma mass spectrometry (ICP-MS) (Supplementary Data 1). Biologically relevant trace elements in the GBS spring water that were absent in the synthetic medium trace metal solution (Supplementary Data 2) included Ni (2.2 nM), which was however in the complete corn stover medium (19–26 nM), and W (263 nM)[23]. Only W was present in the spring water and not elsewhere, such as in the substrate (~0.11 nM) or media components (<0.05 nM, Supplementary Data 2). Furthermore, preliminary analysis of a *W. gerlachensis* MAG obtained from the laboratory cultures indicated the presence of multiple W-associated enzymes and $WO_4^{2-}$ ABC transporters. We therefore hypothesized that W was the key component in the GBS spring water allowing maintenance of *W. gerlachensis*.

The utilization of W was conceivable given its previous recognition as an essential element for diverse ferredoxin oxidoreductases in the marine heterotrophic hyperthermophile *Pyrococcus furiosus* (*Thermococcales*)[24–28]. Nevertheless, W biochemistry is rare in microorganisms, although the low redox potential of W cofactors makes them especially suitable for anaerobes[29]. For instance, the low redox potential of the $W^{6+}/W^{4+}$ couple makes it preferable to Mo, which is more bioavailable in oxic environments and serves equivalent cellular functions in formaldehyde oxidoreductases and benzoyl CoA reductases[30,31]. In acetylene hydratase, W enables sharing of electrons of the outer orbital shell with the substrate and provides electrostatic stabilization to transition states[32]. In GBS, W is likely derived from minerals of the Scheelite ($CaWO_4$) and Wolframite groups ([Fe, Mn]$WO_4$), within W-rich deposits overlaying Triassic clays that can be extensive in the Great Basin[33].

We confirmed that *W. gerlachensis* is dependent on W using a synthetic medium with discrete additions of $WO_4^{2-}$ (Fig. 2). After two transfers in a synthetic medium without W amendment, *W. gerlachensis* fell to very low or undetectable levels based on amplicon sequence variant (ASV) counts, whereas addition of 0.1 or 2 μM $WO_4^{2-}$ alleviated this effect (Fig. 2a). Importantly, the addition of W to the synthetic medium led to *W. gerlachensis* abundances indistinguishable from those in the original spring water-containing medium (ANOVA, $p > 0.05$). These results were also reflected in *W. gerlachensis* absolute abundance as determined by qPCR. The omission of W induced a drop of *W. gerlachensis* 16S rRNA genes by 2–3 orders of magnitude (Fig. 2b). Tungsten starvation induced loss of *W. gerlachensis*

earlier (after the second transfer) in corn stover relative to sugar mix cultures (Fig. 2c and Supplementary Fig. 3), with orders-of-magnitude differences in *W. gerlachensis* 16S rRNA genes with varying W concentrations. Concentrations of 20 nM $WO_4^{2-}$ yielded higher *W. gerlachensis* 16S rRNA gene numbers than 5 nM $WO_4^{2-}$ (corn stover) and 1 nM $WO_4^{2-}$ (sugar mix, ANOVA, $p < 0.05$ for both), suggesting 20 nM $WO_4^{2-}$ was the critical threshold to maintain *W. gerlachensis* over at least three transfers in synthetic medium (Fig. 2c). We should note that, given ~0.8 mM sulfide added to the medium as reducing agent, the dominant W species is likely $WS_4^{2-}$, which microbial cells would be exposed to. Regardless of the W speciation, low W levels decreased the total culture biomass as assessed by DNA concentrations in extracts (Supplementary Fig. 3), suggesting that microorganisms requiring or using W may play important roles in community productivity, despite the relative abundances of only a few community members being directly affected by the absence of W. The critical W concentration to maintain *W. gerlachensis* in culture is higher than that required by *P. furiosus* (at least 1.5 nM but less than 15 nM)[34]. Nevertheless, *P. furiosus* growth dynamics were not determined over multiple transfers. There may be differences in the W-requiring biochemistry between these two organisms, while the comparison between mixed and pure cultures may also be obscured by inter-species W cycling.

Because the abundance of *Caldimicrobium* was also affected by W limitation (Fig. 2a and Supplementary Fig. 4), we were unable to exclude the possibility that *W. gerlachensis* was affected indirectly via *Caldimicrobium*, potentially due to a mutualistic relationship between the two. Nevertheless, the presence of multiple W-associated genes in the *W. gerlachensis* MAG, discussed below, suggested that the requirement for W is direct. To probe the genomic potential for *W. gerlachensis* anaerobic growth on carbohydrates, we annotated a high-quality MAG from a hybrid assembly of short- and long-read sequence data derived from the laboratory cultures (N50 = 86,582 bp, estimated 98.06% completeness and 0.49% contamination, containing both 16S and 23S rRNA genes). This MAG encoded six W-dependent ferredoxin oxidoreductases as well as a complete $WO_4^{2-}$-transmembrane ABC transport system of the Tup family, which corroborated our initial hypothesis that W is required for growth in synthetic medium.

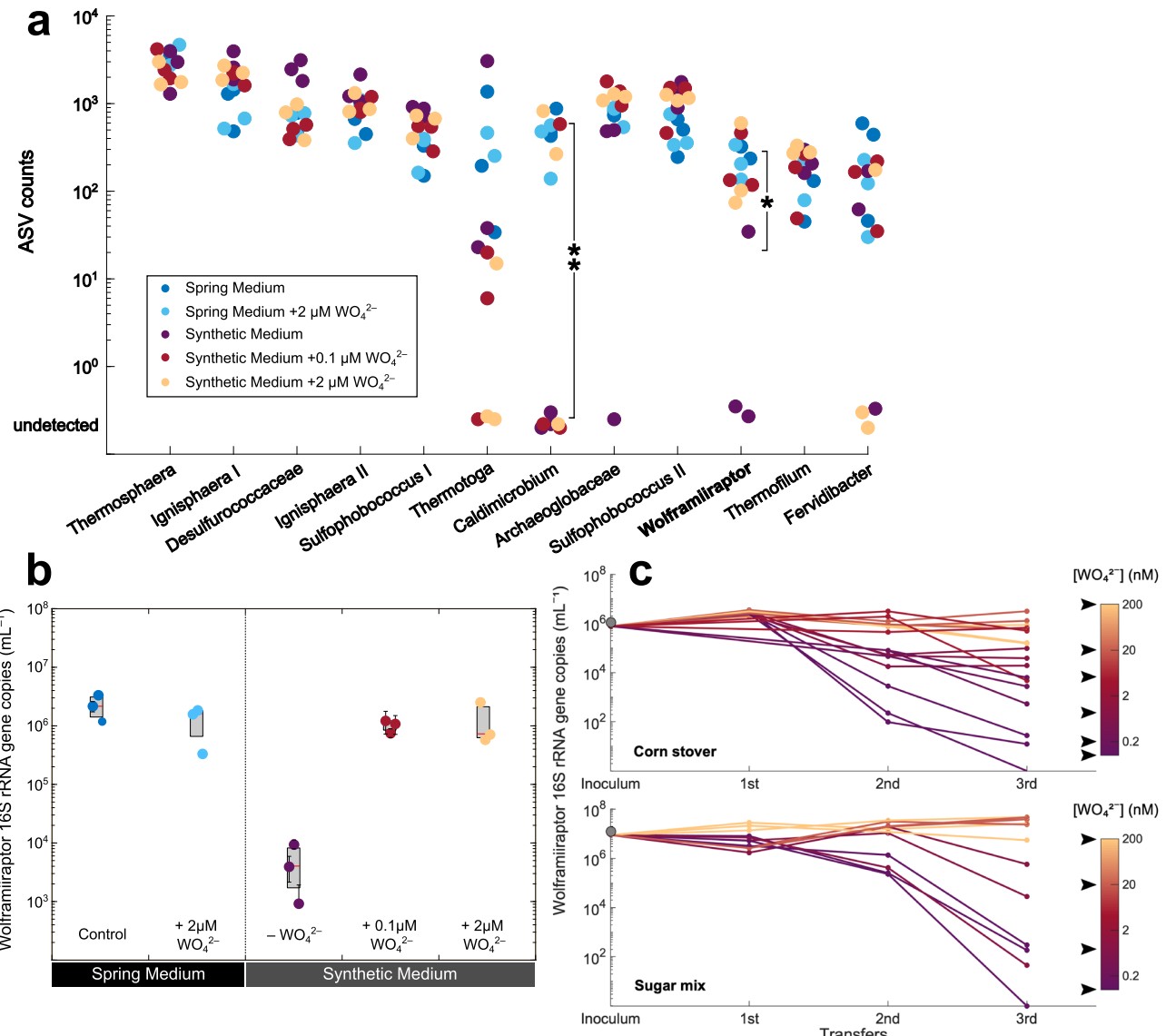

**Fig. 2 Effects of W on *Wolframiiraptor* (*W. gerlachensis*) growth in culture A10. a** ASV abundances based on 16S rRNA gene tags from synthetic medium with and without added spring water or W. The 12 most abundant taxa are shown, which consistently made up >98% of the total ASVs. *Caldimicrobium* and *W. gerlachensis* ASV counts from triplicate cultures were statistically different between synthetic medium with and without spring water (without W addition) as denoted by the asterisk (ANOVA, *$p = 0.07$, **$p = 0.03$, $n = 3$). ASV counts are reported as raw reads; see Supplementary Data 9 including normalized ASV counts that show identical trends. **b** Absolute abundance of *W. gerlachensis* based on 16S rRNA gene qPCR on corn stover incubations. Central marks in boxes indicate the median, with the bottom and top edges as the 25th and 75th percentiles. Error bars denote SD of biological replicates ($n = 3$). **c** Abundance of *W. gerlachensis* 16S rRNA genes measured after each transfer and the resulting growth. Cultures were grown in corn stover and sugar mix in synthetic medium amended with discrete W concentrations (indicated by black arrows). 16S rRNA gene tags retained at the third transfer indicated no ASVs affiliated with *W. gerlachensis* in all replicates without W (Supplementary Fig. 4). Source data are provided as a Source Data file.

W-dependent ferredoxin oxidoreductases are extremely oxygen-sensitive cytoplasmic tungstoenzymes that are involved in sugar and peptide catabolism in a variety of thermophiles. They oxidize various aldehydes, derived from amino acids or sugars, to corresponding carboxylic acids with ferredoxin as the electron acceptor[35]. These enzymes can generally be separated into aldehyde:ferredoxin oxidoreductases (AOR)[24], glyceraldehyde-3-phosphate:ferredoxin oxidoreductases (GAPOR)[25,36,37], and formaldehyde:ferredoxin oxidoreductases (FOR)[26], in addition to tungstoenzymes with unknown function (WOR4)[27] or broad substrate specificity (WOR5)[28]. All characterized enzymes in this family require incorporation of W, which is coordinated by a pyranopterin cofactor, similar to molybdoenzymes[38,39]. Dissolved W is mainly available as the tetrahedral oxyanion

$WO_4^{2-}$ in reduced spring waters[40]. Tungstate is taken up by ABC-type transporters containing highly selective binding proteins (TupA, WtpA). Molybdate is transported by other distinct transmembrane proteins (ModABC). Although we found several annotated $MoO_4^{2-}$ cofactor biosynthesis protein-encoding genes within the *W. gerlachensis* MAG, no Mod genes were present, supporting that multiple cellular processes in *W. gerlachensis* are dependent on W and not Mo.

To test if the identified genes encoding tungstoenzymes and W transporters were expressed, we performed reverse transcriptase-polymerase chain reaction (RT-PCR) assays specific for these *W. gerlachensis* transcripts in community RNA. While several of the oxidoreductase genes were transcribed, $WO_4^{2-}$ ABC-type transporters were only weakly transcribed (Supplementary Fig. 5).

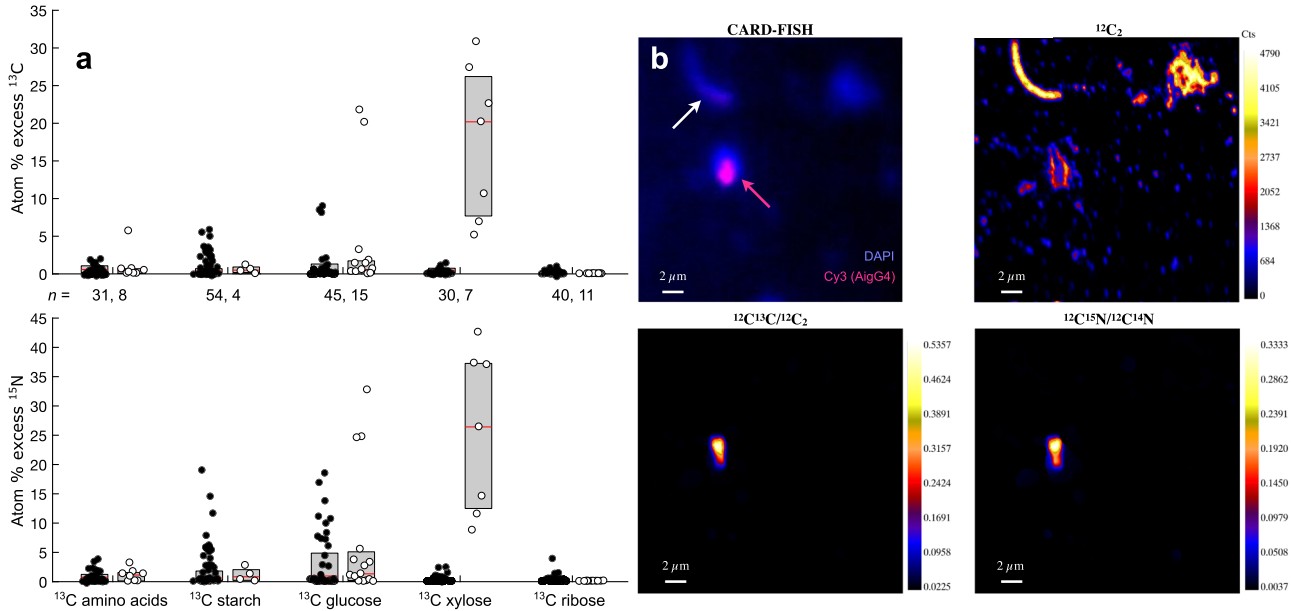

**Fig. 3 $^{13}$C and $^{15}$N incorporation measured by nanoSIMS in single cells from GBS incubations cultured with isotopically labeled substrates. a** Isotopic enrichments in CARD-FISH-identified *W. gerlachensis* (white data points) and other (black data points) cells. Central marks in boxes indicate the median, while the bottom and top edges are the 25th and 75th percentiles. Each point reflects the enrichment of a single cell. **b** Fluorescent (CARD-FISH) and nanoSIMS ion and ion ratio images reflecting $^{13}$C and $^{15}$N incorporation in a representative *W. gerlachensis* cell identified by CARD-FISH in xylose treatments (Cy3, magenta arrow). An adjacent cell (no Cy3 label, white arrow) does not exhibit $^{13}$C and $^{15}$N enrichment. The higher relative enrichment of $^{13}$C and $^{15}$N in *W. gerlachensis* corresponds to higher single-cell anabolic activity compared to other surrounding cells. The minimum isotope abundance ratios are set to natural abundance ratios. Source data are provided as a Source Data file.

Transcripts for the $WO_4^{2-}$-binding portion of the ABC complex (MCF3653660.1) were detected in the sugar mix culture (irrespective of reverse transcriptase), and the transmembrane subunit transcripts were detected in the corn stover culture. Increased synthesis rates of the periplasmic binding proteins over other subunits have previously been observed for ABC-type transporters in *Escherichia coli*, including $MoO_4^{2-}$ transporters, where the synthesis rate of the periplasmic binding protein exceeded that of the transmembrane portion by at least 100-fold[41]. For the putative W-dependent ferredoxin oxidoreductases, the gene encoding protein MCF3653435.1 was transcribed in corn stover and sugar mix cultures, whereas those encoding MCF3653440.1 and MCF3653608.1 were only transcribed in corn stover cultures (Supplementary Fig. 5). The apparent higher expression of tungstoenzymes in corn stover over sugar mix cultures is consistent with higher sensitivities to W starvation (Fig. 2c). Thus, the expression of W-dependent enzymes varied in enrichments provided with different primary carbon substrates, pointing to the direct involvement of tungstoenzymes in the catabolism of complex carbohydrates in corn stover.

**Xylose is the preferred carbohydrate monomer for *W. gerlachensis*.** To investigate carbohydrate metabolism in *W. gerlachensis* in the absence of an axenic culture, we used a combined CARD-FISH-nanoSIMS approach[42] that allowed us to directly interrogate the metabolism of single cells in the complex community of the enrichment. We amended GBS in situ corn stover enrichments with different $^{13}$C-labeled substrates (amino acids, starch, glucose, xylose, and ribose) to track carbohydrate metabolism, and with $^{15}$N-labeled ammonium to track overall assimilatory activity. Cells from non-FISH-labeled incubations collected after 24 h were significantly enriched in both $^{15}$N and $^{13}$C compared to unlabeled controls (Kruskal–Wallis test,

$p < 0.001$; Fig. 3 and Supplementary Fig. 6), indicating active metabolism. Carbon and nitrogen enrichment were positively correlated ($p = 0.003$, Supplementary Fig. 6), but the $^{13}$C/$^{15}$N enrichment varied, suggesting that some cells were metabolically active (labeled with $^{15}$N ammonium) but did not incorporate any of the $^{13}$C-labeled substrates.

The *W. gerlachensis* population identified with CARD-FISH and subsequently analyzed by nanoSIMS was significantly enriched with $^{13}$C and $^{15}$N in all except the ribose-amended incubations (Fig. 3 and Supplementary Fig. 6; Kruskal–Wallis test, $p < 0.001$). *W. gerlachensis* cells were the most active ($^{15}$N enriched) in the xylose treatment. Xylose incubations also contained *W. gerlachensis* cells most enriched in $^{13}$C (median of 15 atom% excess and 100% of cells enriched), consistent with direct assimilation of xylose and a preference for it over the other substrates tested. Median $^{13}$C enrichment of *W. gerlachensis* cells incubated with labeled glucose, amino acids or starch was less than 1 atom% excess (potentially due to cross-feeding on metabolites derived from these substrates), and no isotope enrichment was detected in $^{13}$C ribose incubations.

A hot spring sediment community composed of 4% *Caldarchaeales* degraded the xylose polymer xylan and was used to purify a thermostable xylanase enzyme (>70% activity at 70 °C[43]), indicating the participation of this lineage in polysaccharide-degrading consortia. We also found xylose isomerase (EC 5.3.1.5) and multiple xylulose kinase (EC 2.7.1.17) encoding genes in the *W. gerlachensis* MAG. In light of our NanoSIMS results, these data suggest *W. gerlachensis* is an important contributor to the anaerobic degradation of lignocellulose by the microbial community at GBS, likely as members of a cellulolytic consortium[17]. The precursor of xylose is likely hemicellulose in corn stover, but other sources might be present in natural hot spring environments, including both allochthonous (e.g., nearby plants) and autochthonous (e.g., biofilm) materials.

**W-dependent ferredoxin oxidoreductases putatively drive aldehyde metabolism.** Next, we identified the potential substrates of the six W-requiring ferredoxin oxidoreductases encoded by the *W. gerlachensis* MAG, and likely to be compromised by W limitation, employing primary sequence and structural analyses. Over 2600 sequences of known or annotated AOR, FOR, GAPOR, and WOR homologs served as reference sequences to interrogate the W-associated ferredoxin oxidoreductase homologs encoded in the *W. gerlachensis* MAG. Based on phylogenetic analysis, three homologs were distantly related to well-characterized AORs, herein referred to as AOR-like 1, AOR-like 2, and AOR-like 3; one homolog was distantly related to characterized FORs, referred to herein as FOR-like; and one homolog was distantly related to characterized GAPORs, here termed GAPOR-like (Fig. 4 and Supplementary Figs. 7 and 8). To gain insight into the putative functions of these oxidoreductases, we modeled the protein structure of MAG-encoded oxidoreductase sequences belonging to the genus *Wolframiiraptor*, along with selected reference sequences, and applied multidimensional scaling on the predicted structures (Fig. 4b). Structural similarities to a characterized AOR from *P. furiosus*, as quantified based on a sum-of-pairs score using a protein database reference, emerged in the three AOR-like lineages (Cluster I and II, Fig. 4b), supporting the phylogenetic analysis. The other oxidoreductases were structurally related to the GAPOR from *Caldicellulosiruptor bescii* (*Clostridia*)[37] (Cluster III, Fig. 4b) and diverse FORs (Cluster IV, Fig. 4b), again congruent with the phylogenetic analysis. With the genomic assessments, coupled with the protein phylogenetic-structural analyses, we further looked for evidence of physiological dependence on W.

W-requiring ferredoxin oxidoreductases are found at the heart of sugar and peptide metabolisms in thermophilic anaerobes[44,45]. While glycolysis in thermophilic bacteria is catalyzed by the conventional glyceraldehyde-3-phosphate dehydrogenase and phosphoglycerate kinase, glucose catabolism in several hyperthermophilic archaea differs from that canonical pathway and appears to proceed via a modified Embden–Meyerhof (EM) pathway that involves novel W-dependent enzymes[46]. On one hand, phylogenetic analysis and structural modeling of the homologs represented by MCF3653440.1 in *W. gerlachensis* predicts an encoded GAPOR (Fig. 4, GAPOR-like lineage). On the other hand, the *W. gerlachensis* MAG also encodes an annotated glyceraldehyde-3-phosphate dehydrogenase (EC 1.2.1.12) and a phosphoglycerate kinase (EC 2.7.2.3) and we thus cannot determine with certainty if pyruvate is originating via the canonical or modified (archaea-typical) EM pathway. Similar to the amino acid catabolism in *P. furiosus*, *W. gerlachensis* likely decarboxylates 2-keto acids, such as pyruvate, to aldehydes that are then oxidized by W-dependent ferredoxin oxidoreductases to form carboxylic acids (Fig. 4c)[47–49]. The pivotal step of the metabolic inhibition occurs when W-requiring ferredoxin oxidoreductases become insufficiently supplied with the W-cofactor tungstopterin and ferredoxin reduction halts. A ferredoxin-dependent, membrane-bound proton-translocating [NiFe]-hydrogenase (Mbh) encoded by the *W. gerlachensis* genome would then increasingly lack reduced ferredoxin to trigger tunneling of $H^+$ ions across the cell membrane, thereby leading to a decreased transmembrane $H^+$ gradient and retarded ATP formation (Fig. 4).

Inactivation of W-dependent ferredoxin oxidoreductases would be highly effective at depriving the cell of energy because (i) the encoded ATPase is predicted to be proton-dependent, based on alignment with V-type ATPases[50], inducing a proton gradient of 3 ion equivalents across the membrane and (ii) these oxidoreductases can make up a significant proportion of cell biomass. For instance, in *Thermococcus* strain ES1, these

oxidoreductases constitute 1% of the total protein mass[51] and the intracellular W-dependent ferredoxin oxidoreductase concentration in *P. furiosus* reached 110 μM[44]. Interestingly, the *W. gerlachensis* MAG revealed a putative ferredoxin-dependent, membrane-bound, and proton-translocating [NiFe] hydrogenase complex of 10 subunits (Mbh-Mrp) including conserved cysteine pairs in the N- and C-terminus of the catalytic subunit MbhL (Supplementary Fig. 9) that ligate the [NiFe] metallocluster[52], along with several Hyp-type [NiFe] metallocluster assembly proteins. We were not able to find *rnf* genes[53] in the *W. gerlachensis* MAG and we hence suspect ferredoxin (as the reducing equivalents produced by AOR) would predominantly be recycled by the Mbh-Mrp. A similar Mbh-Mrp complex has previously been described for *P. furiosus*[54,55] and is a type-4 [NiFe] energy-conserving hydrogenase[56] that was also detected in other *Caldarchaeales* genomes[8,12]. W addition into culture media of the marine thermophile *Thermotoga maritima* resulted in stimulation of hydrogenase activity[57]. The carbon and energy metabolism in that bacterium are similar to that of *W. gerlachensis*[58]: Formation of pyruvate via the EM pathway, production of organic acids and $H_2$ and therefore coupling of W enzymatic activity to a hydrogenase and alternative reduction of $S^0$ to $H_2S$ by $H_2$. Across domains and comprising terrestrial as well as marine habitats, W-associated anaerobic heterotrophic processes are therefore important for thermophilic life.

**Distribution of tungstoenzymes across the family *Wolframiiraptoraceae*.** In evaluating the scope of tungstoenzymes in the *Caldarchaeales*, we recruited and annotated 77 additional high-quality MAGs related to *W. gerlachensis* that were extracted from metagenomes from geothermal springs in Tengchong, China; Yellowstone National Park, United States; and Taupo, New Zealand, along with a deep-sea hydrothermal mineral deposit from the Eastern Lau Spreading Center in the Southern Pacific. The assignment of these MAGs to the family *Wolframiiraptoraceae* was defined based on monophyly using conserved marker gene sets and relative evolutionary divergence values. The MAGs could be assigned to four genus-level lineages [including previously recognized AigG4 (syn. EX4484-121, herein *Wolframiiraptor*), AigG5 (syn. JGI-000106-J15, herein *Geocrenenecus*), and AigG7 (syn. NZ13-MG1, herein *Terraquivivens*)[8]], and 11 species (Fig. 5 and Supplementary Fig. 10), based on relative evolutionary divergence (Supplementary Data 3), average amino acid identity, and average nucleotide identity (Supplementary Data 4).

To assess the potential importance of W within the family *Wolframiiraptoraceae*, we interrogated the presence of tungstoenzymes within the genomes of these taxa. The three AOR-like, FOR-like, and GAPOR-like enzymes were widely distributed within the family, each present in more than half of the *Wolframiiraptoraceae* genomes (Fig. 4 and Supplementary Figs. 7 and 8), sparse in other *Caldarchaeales*, and absent from close relatives in the *Nitrososphaeria* (Supplementary Fig. 11). In addition, three divergent putative tungstoenzymes were identified in the *Geocrenenecus* and *Terraquivivens* MAGs.

All protein-coding sequences annotated as the substrate-binding subunits of the Mod (K02020), Tup (K05772), or Wtp (K15495) ABC transporter systems within the family were analyzed to determine the putative substrate specificity of these transporters (Supplementary Fig. 12a, b). All species of the genera *Wolframiiraptor*, *Terraquivivens*, and the sole representative of *Benthortus* encode homologs of the TupABC transporter. Notably, only a single genome of *Geocrenenecus* and members of the species *T. tengchongensis* encoded novel TupA- and/or WtpA-like proteins. The WtpA-like protein has been identified before as a ModA-like homolog in *Pyrobaculum* species[14,59]. The

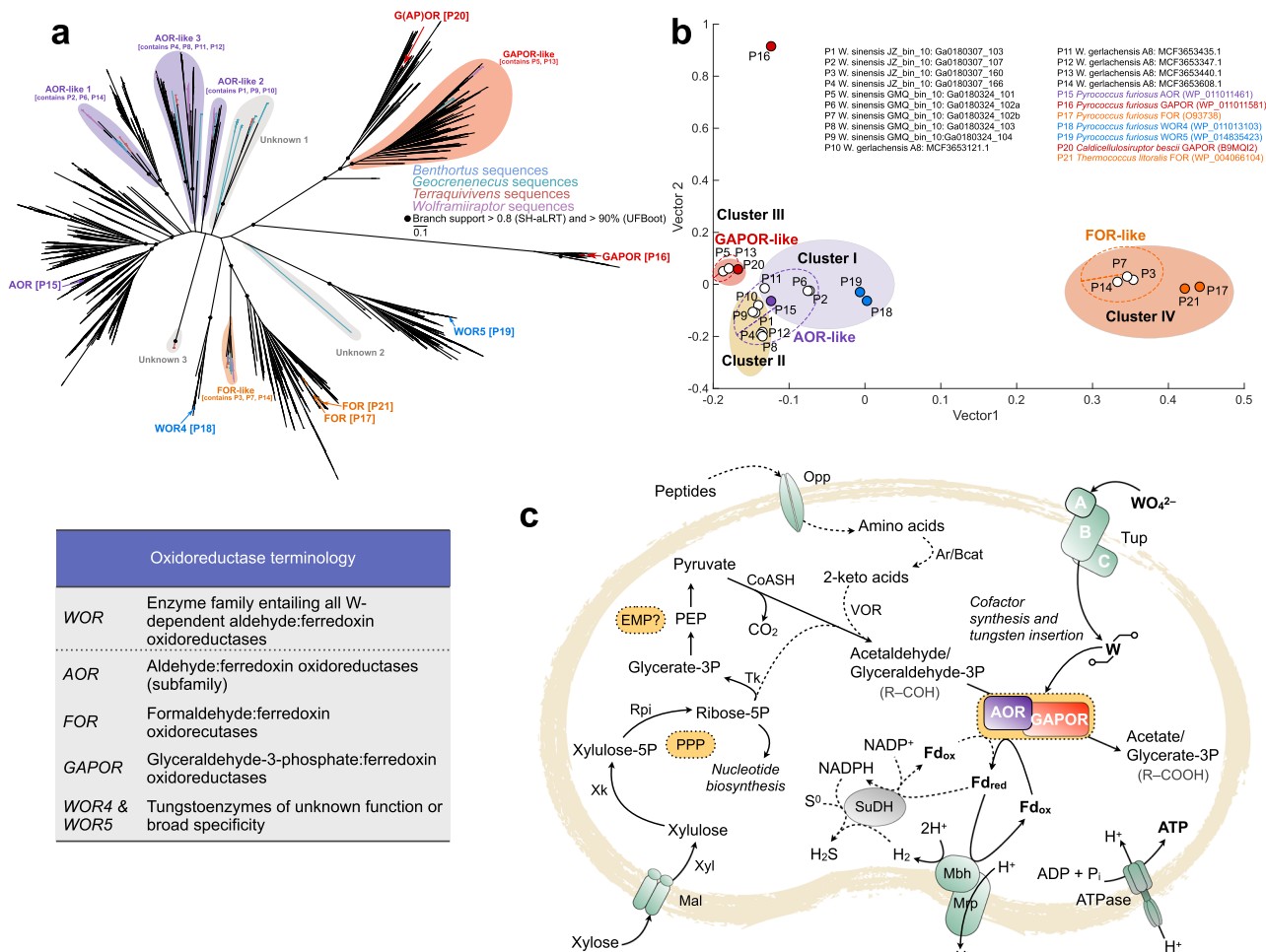

**Fig. 4 Phylogenetic (a) and structural (b) modeling of W-dependent oxidoreductases, and a metabolic model (c) illustrating the proposed link between cellular W usage and energy metabolism in *W. gerlachensis*. a** Maximum-likelihood phylogenetic tree of W-dependent ferredoxin oxidoreductase protein sequences retrieved from MAGs belonging to *Wolframiiraptoraceae*, and reference sequences. Illustrated is the masked alignment of the W-dependent ferredoxin oxidoreductase sequences, excluding all alignment positions with sequence information in less than 75% of the taxa analyzed. Branch support was inferred with SH-aLRT and Ultrafast bootstrapping and support at deeper nodes among the lineages in the phylogeny is indicated. W-dependent ferredoxin oxidoreductase lineages were inferred based on the topologies of the masked and full alignments (Supplementary Fig. 7 and Supplementary Data 7). The scale bar indicates the number of amino acid changes per site. **b** Multidimensional scaling of modeled protein structures. Data points with the most similar structural neighborhoods are positioned near each other. Protein models are based on the amino acid sequences aligned in **a**. Reference sequences are color-coded. K-means clustering was applied to identify significant data clusters (colored ellipses) based on minimum cluster $r^2$. Clusters included AORs from both references and those from representative members of the genus *Wolframiiraptor*. One sequence (MCF3653595.1) was excluded from the analysis (<300 amino acids). Dotted lines are used to indicate putative groups identified in **a** and **b**. All AOR-like lineages grouped together in Clusters I and II, while GAPOR-like sequences grouped in Cluster III and FOR-like sequences grouped in Cluster IV. **c** Pyruvate, derived from xylose via the pentose phosphate pathway (PPP) and the Embden–Meyerhof pathway (EMP), is stepwise converted to organic acids. Within this conversion, AOR and GAPOR couple aldehyde oxidation with ferredoxin reduction. Electrons from ferredoxin are used by a membrane-bound proton-translocating [NiFe]-hydrogenase (*Mbh*) to generate a proton motive force. Besides this main pathway, aldehydes could additionally be derived from amino acid metabolism and ferredoxin could also be recycled by an $H_2$-consuming sulfide dehydrogenase[89] after $H_2$ activation by Mbh (dashed lines). W uptake protein (subunits A-C), *Tup*; Oligopeptide transporter, *Opp*; Aromatic aminotransferase or branched-chain amino acid aminotransferase, Ar/Bcat; 2-Ketoisovalerate ferredoxin:oxidoreductase, VOR; Maltose/maltodextrin transporter, *Mal*; Xylose isomerase, *Xyl*; Xylulose kinase, *Xk*; Ribose 5-phosphate isomerase, *Rpi*; Transketolase, *Tk*; Coenzyme A (-acyl), *CoASH*; Aldehyde:ferredoxin oxidoreductase, *AOR*; Ion translocator, *Mrp*; sulfide dehydrogenase, SuDH; F0F1(V)-type ATP synthase, *ATPase*.

consensus lack of a $MoO_4^{2-}/WO_4^{2-}$ transport system in the *Geocrenenecus* species is not unique, as several completely sequenced genomes from organisms known to utilize $MoO_4^{2-}/WO_4^{2-}$ lack any of these annotated ABC transporters[59].

Scaling up the distribution analysis of $MoO_4^{2-}/WO_4^{2-}$ transport proteins across entire thermophilic community metagenomes indicated unequal *modA*, *tupA*, and *wtpA* gene frequencies among microbial communities inhabiting springs that span a spectrum of geochemical conditions (Supplementary

Figs. 13 and 14). While the $WO_4^{2-}$ transporters show a high presence (~70%) in spring waters with temperatures >60 °C, *wtpA* is generally more abundant than *tupA* and is the most abundant $MoO_4^{2-}/WO_4^{2-}$ transporter among microbial communities of low-pH (<4) springs.

To better understand how *Wolframiiraptoraceae* acquired W-dependent enzymes, we inferred associated evolutionary events through phylogenetic analyses, ancestral character state reconstructions (Supplementary File 2), and gene tree/species tree

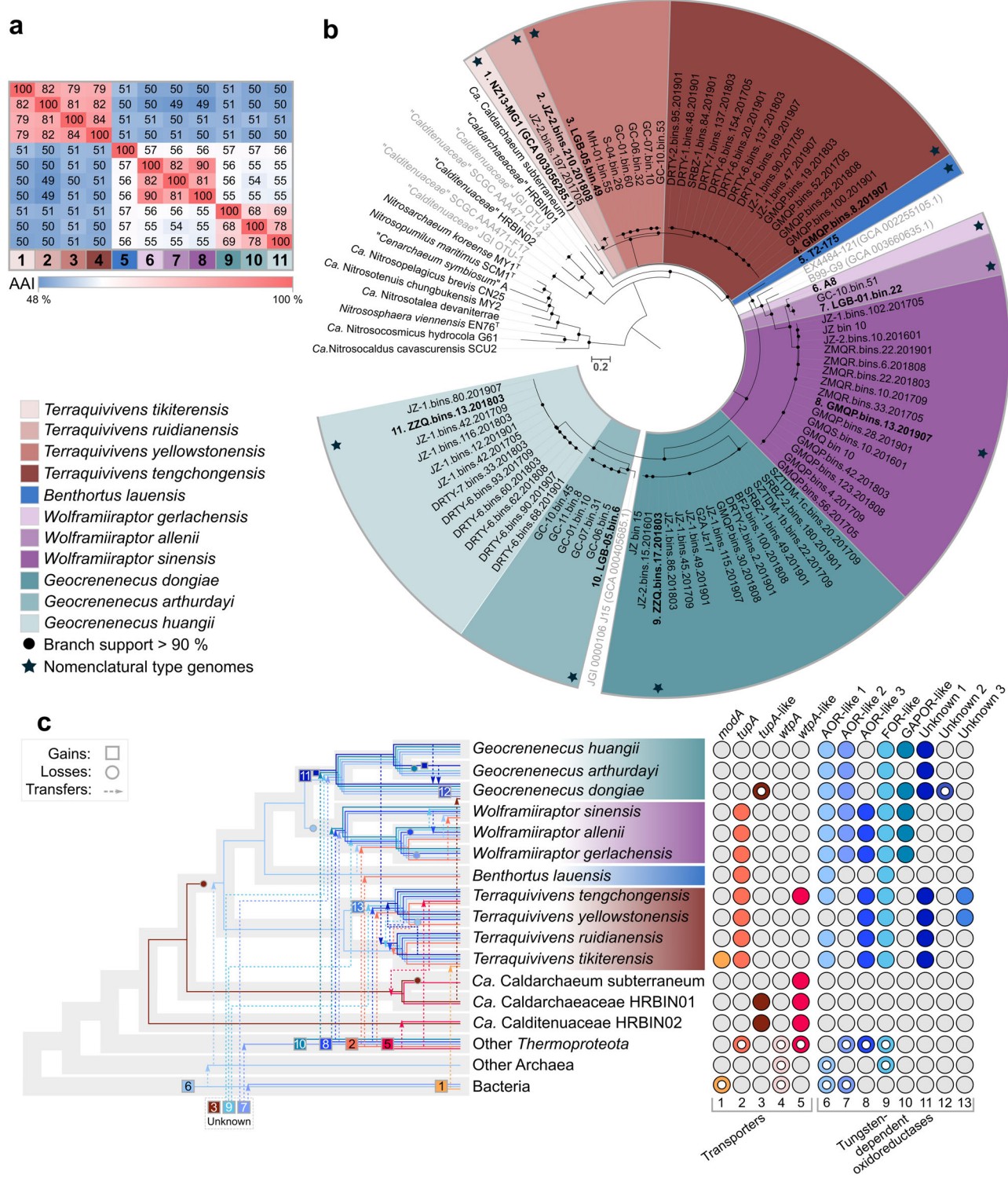

reconciliation analyses based on the W-dependent ferredoxin oxidoreductases and $MoO_4^{2-}/WO_4^{2-}$ transporters (Fig. 5c). These analyses revealed a complex evolutionary history characterized by rampant lateral transfer from other *Thermoproteota* to various *Wolframiiraptoraceae* lineages. Although it remains difficult to distinguish between phylogenetic noise and true signal for lateral acquisitions in single-gene trees, the presence of one tungsten-zyme (AOR-like 1) in the ancestral node to the *Wolframiiraptoraceae* was well supported by all analyses. This ancestral enzyme, combined with extensive capture of W-associated enzymes from

*Thermoproteota* and other sources (Supplementary Figs. 7, 8, and 10 and Supplementary Dataset 2), and multiple gene family expansions and losses throughout the evolution of the *Wolframiiraptoraceae*, implies an important role for W in the diversification of the family. Given the close phylogenetic grouping of the *Wolframiiraptoraceae* TupA transporter subunit, AOR-like 3, FOR-like, and GAPOR-like sequences with those from diverse *Thermoproteota* (Supplementary Figs. 7, 8, and 10 and Supplementary File 3), these enzymes may have been inherited through separate horizontal acquisitions. However, the

**Fig. 5 Genomic relatedness (a), phylogenetic placement (b), and distribution of tungstoenzymes (c) in the family *Wolframiiraptoraceae*. a** Average amino acid identity (AAI) among designated type genomes for species-level groups within the family *Wolframiiraptoraceae*. The same color scheme used in **b** is used for taxa indicated with numbers. For full ANI and AAI matrices of the respective genera, see Supplementary Data 4. **b** A maximum-likelihood phylogeny was inferred from the concatenated, partitioned sequence alignment of 122 conserved archaeal marker sequences (ar122). Appropriate evolutionary models were applied for each partition. Branch support for the phylogeny was inferred from 1000 pseudoreplicates, using traditional bootstrapping and supported branches are indicated with filled circles at nodes. Species-level groups, as supported by phylogenetics, average nucleotide identity (ANI), and relative evolutionary divergence are indicated with different colors, with species belonging to the same genus indicated in different shades of the same color. Taxon names in bold and indicated with a star are the proposed nomenclatural types for the taxa. Gray taxon names represent medium-quality MAGs that are publicly available. **c** Summary of gene tree/species tree reconciliation analyses with predicted gains, transfers, and losses, for W-associated enzymes (ABC-type transporter A subunits and W-dependent oxidoreductases) within the family. Hypothesized evolutionary paths for the genes of interest are indicated in the colors used in the dot matrix on the right of the cladogram and numbered according to the dot matrix order. The consensus presence or absence profiles for species with multiple genome sequences available were determined by data as indicated for the majority of genomes in the species. Where genes encoding these proteins are present in some genomes within a taxon, colored dots are partially filled, while fully filled dots in the dot matrix indicate the presence in the majority of the genomes belonging to that taxon. For full ABC-type transporter phylogeny, including KOs K02020, K05772, and K15495, see Supplementary Figs. 10 and Fig. 5, and Supplementary Fig. 7 for phylogenies and structural modeling of W-dependent ferredoxin oxidoreductases.

notable progressive increases in the number and diversity of W-dependent enzymes within *Caldarchaeales*, and in particular *Wolframiiraptoraceae*, suggest an early emergence of W utilization within the *Thermoproteota* and a critical role for W metabolism in the evolution of the *Wolframiiraptoraceae*.

## Discussion

In the present study, we demonstrate that W is required to culture the anaerobic species *W. gerlachensis*, the first physiologically characterized cultured member of the *Caldarchaeales*. The basis of this requirement, as evidenced by poor biomass yields in cultures in synthetic medium without amended W, is most likely W-dependent ferredoxin oxidoreductases involved in the degradation of xylose-derived pyruvate, which becomes non-functional without W. This dependence on W is seemingly a characteristic of several close relatives of *W. gerlachensis*, with W utilization feasibly representing an ancestral trait within the family and important for the diversification of the *Wolframiiraptoraceae*. The levels of W required by *W. gerlachensis* (≥20 nM) are below those that are typically inhibitory to the growth of other microorganisms where W inhibition has been demonstrated[18]. The standard inclusion of this trace element may aid in the cultivation of additional members of the *Wolframiiraptoraceae* and other novel thermophiles.

It has been suggested that molybdoenzymes did not evolve prior to the widespread oxygenation of the Earth, which allowed for oxidative weathering of sulfide minerals that hosted Mo[60]. In contrast to Mo, W is less affinitive to sulfide complexation and it is possible that the apparent favorability of W over Mo in the *Wolframiiraptoraceae* is a relic from adaptations to early microbial habitats where W was more bioavailable than Mo. On the early Earth, such potential habitats were more common and extended beyond hot springs into the marine realm. About 2.3 billion years ago, increasingly sulfidic bottom waters of the oceans underwent a strong geochemical shift from ferruginous to euxinic[55–57]. Investigation of contemporary euxinic sediments in the Black Sea revealed a critical concentration of ~11 μM aqueous sulfide to cause complete Mo fixation from the water column[61] due to reduction to insoluble Mo-sulfide species or transformation to particle-reactive thiomolybdate ($MoS_4^{2-}$)[62]. Such sulfide levels based on dissimilatory sulfite/sulfate reduction[63] or volcanically derived sources[63,64] likely prevailed both in terrestrial hot springs and marine basins. The availability of W theoretically could have constituted a regulatory switch on the fermentative degradation of organic matter and could have influenced fluxes of organic acids, which, in turn, were crucial substrates for sulfur reduction and methanogenesis. The selective pressure for W utilization likely dropped with the oxygenation of the Earth and the dispersion of soluble Mo, but the remnants of ancient W-based pathways likely perpetuated in the carbohydrate metabolisms of anaerobic, thermophilic archaea.

## Methods

**In-situ enrichments and sampling**. GBS water used in the preparation of cultivation medium was sampled in 2013, 2016, 2017, and 2018, passed through a 0.2 μm filter, and stored in 20 L plastic containers in the laboratory without temperature control. In situ enrichments on ammonia-fiber explosion-treated corn stover (kindly provided by Bruce Dale, Michigan State University) in GBS were performed as described in Peacock et al.[17]. Samples were taken after approximately six months of incubation on March 29, 2016. At that time, the temperature of the water was 86.0 °C and the pH was 7.15, typical for the main source pool of GBS[16]. For sampling, a nylon bag containing corn stover was removed from the conical tube, the bag was cut open with a sterile scissors, and approximately half of the contents was immediately transferred to 10 mL of 0.2 μm filtered GBS water (previously sampled, sparged with $N_2$, reduced by the addition of cysteine to 0.05 g/L, and sterilized by autoclaving) in a 25 mL Balch tube under a stream of $N_2$ gas applied through a 0.2 μm filter. The tube was sealed with a butyl rubber stopper, the headspace was flushed for 3 min with $N_2$ gas, and the tube was then shaken to disperse the corn stover. Previously prepared anaerobic media (50 mL in 160 mL serum vials sealed with butyl rubber stoppers; see below) were inoculated with 0.2 mL of the corn stover slurry using a sterile, $N_2$-flushed needle and syringe. The bottles were transported back to the laboratory at ambient temperature and were transferred to high-temperature incubators within 24 h of sample collection. The remaining in-situ corn stover enrichment was placed in 1.5 mL centrifuge tubes, frozen immediately, and transported back to the laboratory on dry ice, and then stored at −80 °C for DNA extraction.

**Laboratory culturing**. The 13 different media and incubation conditions used are shown in Supplementary Table 1. All media were based on GBS salts synthetic medium, which contained per liter of 18.2 MΩ cm water: 3 g NaCl, 0.3 g $Na_2SO_4$, 0.15 g KCl, 0.015 g $CaCl_2 \cdot 2H_2O$, and 0.123 g $MgSO_4 \cdot 7H_2O$, and 5 mL of a trace mineral solution[64]. The trace mineral solution, based on ref. [65], contained per liter: 3.7 g $Na_2$-EDTA, 1.1 g $FeSO_4 \cdot 7H_2O$, 0.028 g $ZnSO_4 \cdot 7H_2O$, 0.098 g $MnCl_2 \cdot 4H_2O$, 0.031 g $H_3BO_3$, 0.024 g $CoCl_2 \cdot 6H_2O$, 0.017 g $CuCl_2 \cdot 2H_2O$, and 0.024 g $NaMoO_4 \cdot 2H_2O$. The "spring water medium" consisted of a 1:1 mix of the above synthetic medium and water collected from GBS as described above. All media contained 0.107 g/L $NH_4Cl$. The medium was then sparged with $N_2$ gas and transferred into serum vials in an anaerobic chamber. The vials were sealed with butyl rubber stoppers and aluminum crimps, the headspace was exchanged with $N_2$ by three cycles of vacuuming to 585 mm Hg and repressurizing to 15 psi with $N_2$, and the vials were sterilized by autoclaving[64]. Prior to inoculation, the following were added to all media at the indicated final concentrations from concentrated, anoxic stocks: 0.025% keratin, 0.002% xyloglucan, 0.02% $Na_2S$, 0.02% cysteine, 0.002% yeast extract, 0.0014% $NaH_2PO_4 \cdot H_2O$, and 1.6 mL/L of a vitamin solution[66]. All stocks were sterilized by autoclaving except yeast extract and vitamins, which were passaged via a 0.2 μm filter. Additions for different media conditions in the initial enrichments were made from sterile, anoxic stocks to final concentrations shown in Supplementary Table 1, except that corn stover (0.02% m/v) was added to individual serum vials in the anaerobic chamber prior to sealing. Follow-up experiments used the same base synthetic or spring water media with modifications of the specific substrates that were excluded or added anoxically. After stable laboratory cultures were obtained, the culture conditions for routine transfer and maintenance used the base spring water medium with 2 mM sodium bicarbonate and either corn

stover (0.02% m/v) or a "sugar mix" (0.01% m/v each of glucose, xylose, mannose, and D- and L-arabinose) for major carbon sources. Laboratory cultures were kept at 80 °C and initially transferred to fresh media (1/100 volume) every 2 weeks for the first two transfers after inoculation with in situ enrichments and then every 3 weeks thereafter.

**Quantification of W in spring water and media by ICP-MS and ICP-OES**. W levels in laboratory media and filtered GBS spring water were measured using ICP-MS after the addition of nitric acid (OPTIMA grade, Fisher Chemical, Fair Lawn, NJ, USA) to 1% final concentration at Huffman Hazen Laboratories (Golden, CO, USA). W levels in filtered GBS spring water were also measured using inductively coupled plasma optical emission spectroscopy (ICP-OES) with an Agilent 5110 ICP-OES (Santa Clara, CA, USA) and dilutions of a W calibration standard (PLW9-2Y, SPEX CertiPrep, Metuchen, NJ, USA). Nitric acid was added to samples and standards to a final concentration of 1% before analysis, and optical emission at 207.91 nm was used for W quantification. Broad trace element analysis by ICP-MS on GBS spring water and media was performed on samples prepared as described above at the Analytical Chemistry Laboratory of New Mexico Bureau of Geology and Mineral Resources, New Mexico Institute of Mining and Technology.

**16S rRNA gene tag sequencing**. 16S rRNA gene tag amplification and sequencing on initial enrichment cultures was performed essentially as described[67]. A modified forward primer 515 (5′ GTGYCAGCMGCCGCGGTAA) with a Y instead of a C at the fourth position from the 5′ end was used to increase coverage of Archaea, and a corresponding change was made to the SeqRead1 primer. Sequencing (2 × 250 bp) was performed on an Illumina iSeq sequencer using the V2 reagent kit. Additional 16S rRNA gene tag sequencing on laboratory cultures tested with different levels of W was performed as described[68] at Argonne National Laboratories.

**Genomic DNA and RNA extraction**. For the laboratory culture, biomass (from 10 to 40 mL culture) was harvested by centrifugation at 10 min at 9300 g and 4 °C in 50 mL conical tubes. Resulting pellets were resuspended in 1 mL of supernatant, transferred to 1.5 mL tubes, and centrifuged at 16,100 g for 5 min at 4 °C. The supernatant was removed and the pellets were stored at −80 °C until nucleic acid extraction. DNA was extracted using the Fast DNA SPIN Kit for Soil (MP Biomedicals) as described following the manufacturer's recommendations, except that homogenization by bead-beating was performed twice for 30 s at a speed setting of 4.5 m/s. DNA was eluted in 75–100 µL of DES solution and further diluted 10-fold in 10 mM trishydroxyaminomethane pH 8. To obtain long fragments of DNA for Oxford Nanopore sequencing, DNA was extracted using the cetyltrimethyl ammonium bromide protocol of the Joint Genome Institute[66]. RNA was extracted using the FastRNA Spin Kit for Soil (MP Biomedicals) and the High Pure RNA Isolation Kit (Roche) as described in Hamilton et al.[69], followed by an additional DNase treatment using the TURBO DNase kit (Ambion) according to the manufacturer's instructions. For all other information regarding metagenomic DNA extraction, please refer to Supplementary Note 1.

**Quantitative PCR**. Assays were performed in 96-well plates (Applied Biosystems) in a StepOnePlus Quantitative Real-Time PCR machine (Applied Biosystems) using PowerUp SYBR Green Master Mix (Applied Biosystems). Cycling conditions for AigG4-specific primers (Supplementary Table 2) were as follows: initial denaturation for 15 min at 95 °C followed by 45 cycles of denaturation (20 s at 95 °C), annealing (25 s at 64 °C), and extension (50 s at 72 °C). Specificity of qPCR primers was verified by melt curve analysis. Using a plasmid with a fragment of the AigG4 16S SSU rDNA sequence as a standard (SSW_L4_C03; ref. [23]), an AigG4 standard curve was generated over 6 orders of magnitude from $5.0 \times 10^2$ to $5.0 \times 10^8$ ($r^2 = 0.98$–1.00). The reported template abundances are the average and standard deviation of qPCR assays performed in triplicate. Quantification of total bacterial and archaeal 16S rRNA genes was performed similarly using primers 806r and 515f with the modification described above for 16S rRNA gene tag sequencing (Supplementary Table 2), and with an annealing temperature of 55 °C.

**Reverse transcriptase PCR**. RNA was converted to cDNA with random hexamer primers using the Verso cDNA kit (ThermoScientific), including control reactions without added reverse transcriptase. cDNA was amplified with primers targeting AigG4 predicted W-associated genes and AigG4 16S rRNA (Supplementary Table 3) in the Axygen® Maxygene™ II Thermal Cycler as follows: at 95 °C for 3 min followed by 35 cycles of denaturation (30 s at 95 °C), annealing (30 s at annealing temperature appropriate for specific primer set), and extension (72 °C for 5 min) and finally, 72 °C for 1 min. Primers targeting AigG4 predicted W-associated enzymes were designed using Primer3[65]. DNA was quantified using the Qubit 4.0 with HS DNA reagent kit.

**Stable isotope labeling**. Stable isotope labeling was performed on-site at GBS in July 2017 using in situ corn stover enrichments prepared and incubated as described above but using 5 g packets of corn stover incubated at the GBS "C" site (outflow)[16] for 34 days. Two packets (approximately 10 g) of freshly harvested corn stover enrichment were added to 180 mL of anaerobic, autoclaved GBS water in a

sealed 500 mL bottle with an $N_2$ headspace and shaken to form a slurry. Samples of this slurry (2 mL) were then added via syringe to sealed 60 mL amber serum vials containing 8 mL of anaerobic, autoclaved GBS water with an $N_2$ headspace, generating 10 mL microcosms. Various individual $^{13}$C-labeled substrates (Cambridge Isotope Laboratories) were then added from concentrated, anaerobic stocks to the microcosms at 0.001% (mass/volume) final concentration: algal amino acids, algal starch, glucose, xylose, ribose, or an equal-mass mixture of acetate, propionate, and butyrate; a control with no label added was also performed and did not yield enriched cells. In addition, 0.5 mM $^{15}$N-NH$_4$Cl was added to all microcosms. Preparation of microcosms was done in a hot water bath at 75 °C to prevent excessive cooling. Unlabeled controls were processed as described below immediately after preparation. Sealed microcosms with labeled substrates were incubated at 75 °C for 4 or 24 h. After incubation, microcosm vials were immersed in ice water to cool, shaken vigorously for 30 s, opened, and decanted into 15 mL conical tubes. Tubes were centrifuged for 30 s at 11 g in a clinical centrifuge to pellet large fragments of corn stover. Supernatant (4.5 mL) was then split between three 1.5 mL tubes and centrifuged for 5 min at 16,100 g to pellet cells. Pellets from the three tubes were pooled in 0.25 mL of 1×PBS, and 0.5 mL of 3% paraformaldehyde (PFA) was added and mixed. Samples were fixed for 1 h on ice. After fixation, tubes were centrifuged for 5 min at 16,100 g. Resulting pellets were washed twice with 1×PBS, resuspended in 200 µL of 50% ethanol, and stored on ice (in the field) and then at −20 °C in the laboratory. Prior to performing FISH, Nycodenz density cushion centrifugation[70] was used to separate cells from higher density particles in the samples. Cells harvested from the aqueous-Nycodenz interface were pelleted by centrifugation, washed twice in water, and resuspended in 50% ethanol.

**CARD-FISH**. Detection of *W. gerlachensis* (AigG4) cells was performed using CARD-FISH using a horseradish peroxidase-labeled probe 5′ CRGAAAGGCCTT CAACCTGT (Biomers.net, Germany). This probe is specific to Aigarchaeota Group 4 and Group 5, and was designed based on multiple sequence alignment of 16S rRNA genes of Aigarchaeota and other Archaea and checked using the Probe Match function of the Ribosomal Database Project[16]. Optimization of formamide concentrations used during hybridization was performed using the Clone-FISH technique[70] with the AigG4 16S rRNA gene sequence SSW_L4_C03 (EU635908)[23] expressed from the T7 promoter on plasmid pET23(+) in *E. coli* strain JM109(DE3). Formamide concentrations from 0 to 50% were tested in 10% increments. The highest formamide concentration that gave strong signal (20%) was then used to test for signal in the *E. coli* strain without induction of expression of the AigG4 16S rRNA as well as *Staphylococcus aureus* and *Bacillus subtilis* as negative controls. CARD-FISH was performed as previously described[71] with minor modifications. Briefly, cells were fixed using a 1% PFA solution for 1 h, followed by storage at −20 °C until use. The protocol was performed on fixed cells spotted and air dried on custom synthesized 14-well indium tin oxide (ITO)-coated slides rather than using filters. Cells were permeabilized by treatment with 10 mg/mL lysozyme (for Clone-FISH in *E. coli* and negative controls) or 1 µg/mL proteinase K (for samples with AigG4) in 50 mM EDTA and 100 mM Tris (pH 8) for 60 min at 37 °C, endogenous peroxidases were inactivated by treatment with 0.01 M HCl for 15 min at ambient temperature (~25 °C), and dehydrated using 96% EtOH. Hybridization was performed at 20% formamide for 16 h at 46 °C, followed by washing at 48 °C, CARD with tyramide labeled with Alexafluor 555 (Invitrogen), and counterstaining by incubation in ice-cold 1 ug/mL DAPI for 10 s. Cells were visualized with a Leica DM5500B microscope with a ×100 magnification dry immersion objective. Fluorescence and brightfield images were collected at diverse locations with FISH-positive cells and the X-Y location of those images was noted, as well as fiducial locations to enable navigation in the nanoSIMS.

**NanoSIMS**. ITO-coated slides were cut with a diamond saw in order to fit on metal sample holders and analyzed on a CAMECA nanoSIMS 50 at Lawrence Livermore National Laboratory. The charge-coupled device camera was used to find fiducial locations and locations of the FISH-positive cells using X-Y coordinates and the real-time imaging unit. The primary $^{133}$Cs$^+$ ion beam was set to 1.5 pA, corresponding to an approximately 100 nm diameter beam diameter at 16 keV. Rastering was performed over 20 × 20 µm analysis areas with a dwell time of 1 ms pixel$^{-1}$ for 19–30 scans (cycles) and generated images containing 256 × 256 pixels. Sputtering equilibrium at each analysis area was achieved with an initial beam current of 90 pA to a depth of ~10 nm, thus ensuring analysis of intracellular isotopic material but without sputtering through the ITO coating. After tuning the secondary ion mass spectrometer for mass resolving power of ~7000, secondary electron images and quantitative secondary ion images were simultaneously collected for $^{12}$C$_2^-$, $^{13}$C$^{12}$C$^-$, $^{12}$C$^{14}$N$^-$, and $^{12}$C$^{15}$N$^-$ on individual electron multipliers in pulse counting mode. All nanoSIMS datasets from two independent experimental runs were initially processed using L'Image (http://limagesoftware.net, August 2021) to perform deadtime and image shift correction of ion image data before creating $^{13}$C$^{12}$C/$^{12}$C$_2$ and $^{12}$C$^{14}$N/$^{12}$C$^{15}$N ratio images, which reflected the level of $^{13}$C and $^{15}$N incorporation into biomass. Regions of interest for isotopic ratio quantification were drawn manually around each cell. A cell was considered isotopically enriched, and therefore anabolically active, if the isotopic composition exceeded three times the standard deviation of the mean isotopic composition of unlabeled (non-AigG4) cells[72].

**Metagenome sequencing, assembly, and binning.** For the AigG4 cultures established from the in-situ enrichment at GBS, extracted DNA were sequenced using the Illumina MiSeq platform (2 × 250 bp), along with Oxford Nanopore sequencing using a MinION device. All extracted metagenomic DNA from geothermal springs from Tengchong, China, and Yellowstone National Park, USA, and the marine hydrothermal vent in the Western Pacific were sequenced on the Illumina NovaSeq, HiSeq3000, or HiSeq4000 platforms, in a paired-end or single-read configuration (Supplementary Note 2). Quality control of sequencing reads are outlined in the Supplementary Notes. Sequencing data were assembled and metagenomes were binned using several software packages (Supplementary Note 2). All MAGs were subjected to quality checks with CheckM v. 1.0.18-1.1.3[73] and classification with GTDB Toolkit v. 1.4.1[74]. This was followed by gene calling using Prodigal v. 2.6.2[75] and annotation of all MAGs using eggNOG-mapper v. 2[76], and gene calling and annotation with Rapid Annotation using Subsystem Technology[77] in the case of the hybrid assembly for *W. gerlachensis*.

**Phylogenomic analyses.** All MAGs classified as belonging to the family NZ13-MG1 in the order *Caldarchaeales* were used for the construction of a species tree. As reference sequences, genomes for members of the *Nitrososphaerales* (syn. Thaumarchaeota), along with publicly available genomes for the *Caldarchaeales* (syn. Aigarchaeota) were included in the taxon set. The 122 conserved archaeal marker sequences in the ar122 dataset were identified from the genomes of these taxa using the GTDB Toolkit v. 1.4.1[74]. Aligned markers were subjected to testing for appropriate evolutionary models using ProtTest v. 3.4[78], followed by concatenation and partitioning with FASconCAT-G v. 1.0.4[79]. This partitioned, concatenated matrix was subjected to maximum-likelihood tree reconstruction with RAxML v. 8.2.12[80], with the appropriate evolutionary model applied to each partition, and branch support inferred from 1000 bootstrap pseudoreplicates. For description on the phylogenomic analysis for the domain Archaea, please see Supplementary Notes 3–6.

**Phylogenetic analyses of genes.** For phylogenetic confirmation of function of the predicted genes related to physiology, and specifically tungstoenzymes, all *Wolframiiraptoraceae* sequences annotated as belonging to the KEGG Orthology (KO) numbers of interest were used to infer phylogenetic trees to place the identified sequences into evolutionary context within the protein families. A detailed description of these analyses is provided in the Supplementary notes (see also Supplementary Data 7 and 8).

**Gene tree/species tree reconciliation and ancestral character state reconstruction.** For reconciliation analyses, a species tree was inferred from the domain-level phylogenomic tree with Bacteria specified as the outgroup. Similarly, subtrees for all W-associated homologs were inferred from the maximum-likelihood phylogenetic analyses of single-gene trees (see Supplementary materials for full description of single-gene tree construction). The inferred gene trees were reconciled with the species tree for the family ecceTERA v. 1.2.4[81], with default reconciliation penalties, partially ultrametric trees specified, and the median reconciliation graphs printed as sylvx reconciliations. Resulting sylvx reconciliations were viewed in SylvX v. 42[82], and median symmetrical reconciliations were summarized and edited in Inkscape v. 0.92.4.

A cladogram inferred from the species tree constructed from the genomes was used as the basis for performing reconstruction of ancestral character states for the family, with a particular focus on genes relating to physiology and W utilization. As the presence or absence of genes within single MAGs can be unreliable, a majority-rule consensus approach was employed for all genomes of the respective species. This entailed using all eggNOG annotated genomes, and summarizing the genome content based on KO numbers for each genome. The absence (0), presence (1), or presence in multiple copies (≥2) for each species was determined based on all genomes belonging to the species. These consensus presence/absence profiles for the species were then used to reconstruct all potential evolutionary events (gene birth, HGT, duplication, or loss) by optimizing rates and inferring the history of gene families by posterior probabilities, as implemented in COUNT[83]. The COUNT sessions for the full genome analysis and the tungstoenzyme-specific analysis are available as Supplementary Data 6.

**Protein structure modeling.** W-dependent oxidoreductase sequences were submitted to the iterative threading assembly refinement (I-TASSER) server[84] to generate models of oxidoreductase homologs. Models were selected based on C-score[85]. All *Wolframiiraptor* W-dependent oxidoreductases reached at least a C-score of 1.3. Structural comparison of all-against-all was done with DaliLite v5 based on PDB coordinates of the models[86,87]. The first two eigenvectors of the output were used for the multidimensional-scaling and K-means clustering[88] applied in JMP Pro (Version 13.1.0, SAS Institute Inc.).

**Reporting summary.** Further information on research design is available in the Nature Research Reporting Summary linked to this article.

## Data availability

Sequence data for nomenclatural types and some non-type genomes were deposited on GenBank under accession numbers listed in Supplementary Data 3, and all other genomes were submitted to eLMSG and/or IMG under under Bioproject nos. PRJNA495050, PRJNA791658, PRJNA807863, PRJNA381623, and PRJNA780009. For nomenclatural types, raw sequence reads were submitted to the Sequence Read Archive (SRA) under the SRA run accessions listed in Supplementary Data 3. All other data are contained within the manuscript and supplements/source data files. Source Data are provided with this paper.

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

## Acknowledgements

We thank Jonathan Covington (UNLV) for help assembling a repository of tungstoenzymes, Jim Noblet and Andreas Beyersdorf (CSUSB) for assistance with ICP-OES, Dave and Sandy Jamieson for access to GBS, high school teachers Kathleen Diver, Heather Garcia, Juan Salgado, and Chuong Vu for assistance with fieldwork, and Prof. Aharon Oren for assistance on nomenclature. We acknowledge Tikitere Trust for their enthusiasm for our research and assistance in access and sampling of the Hell's Gate geothermal features. We extend our gratitude to Mackenzie Lynes and Roland Hatzenpichler for permitting use of metagenome 3300028735 and providing geochemistry information on hot spring LCB-006, as well as to Donald Bryant, Vera Thiel, Mircea Podar, Susannah Tringe, Robert Kelly, Brian Yu, and William Inskeep for sequence data use permissions. We also thank Everett Shock for discussions on W geochemistry in hot springs. Part of this work was carried out at Lawrence Livermore National Laboratory (LLNL) under Contract DE-AC52-07NA2734. Funding was provided by the U.S. National Science Foundation (DEB 1557042 and EAR-1820658), NASA (80NNSC17KO548 and 80NSSC19M0150), and the National Natural Science Foundation of China (No. 91951205).

## Author contributions

J.A.D., B.P.H., A.E.D., S.B. and M.P. conceived of the study. J.D., M.M., C.G., L.G., C.S., C.V., M.Z., N.B., and F.R. conducted cultivation experiments, monitoring, and quantitative PCR. S.B. and J.A.D. analyzed qPCR and 16S data. D.M. performed CARD-FISH. X.M., P.K.W., and J.P.-R. analyzed culture samples with nanoSIMS and S.B. and A.E.D. joined for the nanoSIMS data analysis/interpretation. J.-Y.J., D.R.C., L.M.K., E.S.J., E.S.B., A.D.C., B.S.C., Z.-S.H., W.-J.L., A.-L.R., and M.B.S. were involved in metagenomic sampling, sequencing, assembly, annotation and binning. M.P. performed phylogenomic analyses and ACS reconstructions. S.B. carried out the protein structural modeling. S.B., M.P., and J.A.D. wrote the manuscript and all authors contributed to the final version of the paper.

## Competing interests

The authors declare no competing interests.
