## [Peer Review File · Nature Communications]

Reviewers' Comments:

Reviewer #1:

Remarks to the Author:

Comments on Buessecker et al. "An essential role for tungsten in the ecology and evolution of a novel lineage of anaerobic, thermophilic Archaea"

I am not specialized in microbial cultivation techniques and have not worked with stable isotopes, nanoSIMS and CARD-FISH. I have experience with metabolic pathway analyses with aerobic bacteria but am not familiar with anaerobes and archaea. So I am not qualified to comment on these aspects. I do have rich experience with microbial evolutionary genomics, so my detailed suggestions are limited to these parts.

The evolutionary genomic analyses are not solid, nor complete, and the evolutionary story inferred from those analyses is not supported. There are a lot of discussion on the benefit of using W over Mo as a metal cofactor for microbes living in the early anoxic Earth before the GOE (2.33 Ga ago). These discussions are important and interesting. Without these, the part left is just the finding of W requirement by *Wolframiraptor gerlachensis*, and the potential (not fully consolidated) biochemical mechanism underlying this W requirement, which is important but represents an incremental advancement, certainly not sufficient to make a very attractive story. Note that obtaining the enrichment culture of *Wolframiraptor gerlachensis* is not new, as similar thing of a sister group has already been reported, which was noticed by the authors.

The deep time evolution story that the authors tried to present and highlight builds on two key assumptions: (1) the last common ancestor (LCA) of *Wolframiraptoraceae* used W, and (2) the LCA evolved before GOE. There are a number of important problems. Apparently, testing the assumption (2) requires estimating the age of *Wolframiraptoraceae* (the crown group of *Wolframiraptoraceae*), which is completely missing. This is possible, though few fossils are available in this lineage or even in Archaea. For example, the authors may consider building a tree that include archaea and eukaryotes, and use the rich fossils contained in eukaryotic lineages to help estimate the age of *Wolframiraptoraceae*. This of course requires additional expertise, but again this analysis is important and feasible. For testing the assumption (1), the authors employed COUNT. This is far from being sufficient. There have been a lot of concerns regarding birth-and-death models, based on which COUNT was developed, to infer the gene gain and loss events along an evolutionary tree. It's essential to try a completely different type of methods, in particular those based on gene tree species tree reconciliation. The birth-and-death models and the tree topological incongruence based methods often give different results. The authors can be confident on the inferred events supported by both types of methods. For any inconsistent predictions, additional analyses must be followed to convince people if one prediction is favored over the other by the authors.

Another problem of this manuscript is its flow. One reason is that it contains many taxonomic names. It's difficult for microbiologists who don't work on these groups to follow the paper. It's even more difficult for people working in other areas.

Reviewer #2:

Remarks to the Author:

The authors successfully revealed the carbon and energy metabolisms of a previously uncultivated lineage of *Caldarchaeales*, and the impacts of tungsten on them. Moreover, the authors discussed about the importance of tungsten on the early life on Earth, and the historical importance of the metabolisms related to tungsten in thermophilic life inhabiting geothermal and hydrothermal ecosystems. The carbon and energy metabolisms of the archaeon was unveiled by holistic approach applicable for uncultivated microorganisms. The experimental design and manuscript are generally well organized.

However, very important previous reports presenting the significance of tungsten on carbon and energy metabolisms in hyperthermophilic archaeon *Pyrobaculum* are absent (e.g. Hagedoorn et al. 2005; de Vries et al. 2010, Braseb et al. 2014 and Habib and Hoffman 2017). *Pyrobaculum* is one of the most predominant hyperthermophilic archaeon in high temperature geothermal and coastal

hydrothermal ecosystems with neutral or slightly acidic pH. Thus, the discussion related to evolution and early life in this study might be influenced by these studies. (Note that *Pyrobaculum* has been recognized as a member of Crenarchaeota that is a sister clade of Aigarchaeota. In the novel taxonomic scheme (Rinke et al. 2021) used in this study, Crenarchaeota is classified as Thermoproteia.) In addition, I would suggest to compare the carbon and energy metabolisms in other (hyper-)thermophilic organisms such as Thermococcales and Thermotoga. The linkage between carbohydrate degradation and ATP generation systems of these organisms are similar to *Wolframiraptor*. Thus, such comparison would be beneficial in the discussion of the evolution of early thermophilic life and metabolisms related to tungsten.

L63: "fermentation" has versatile meanings, and is confusing for readers considering the foci of the manuscript.

L76: See the major comments.

L78: Euryarchaeota is not applicable in the taxonomy used in this study.

Fig 1: Please briefly present the methods used to obtain absolute abundance of both total microbial communities and *W. gerlachensis* more clearly. In M&M and supplementary material, I could not identify the method to obtain total microbial abundance.

L159: hyperthermophile

L195: "difference of the systems that require W" would also explain.

Fig.2: I think normalization is necessary to compare ASV abundance but not mentioned.

L621- and 631-: Information about primer sets is missing.

L625: Did the authors confirmed the amplicon by gel electrophoresis?

L731: Delete Ca.

L767: Please complete the data deposition prior to the revision.

Supplementary Text

P12: What does "Sulfide-dependent respiration" mean?

P13: Pacific Ocean

It is not necessary to present accurate sequence length of incomplete binned genome because the lengths must be influenced by parameters in assembly and binning. Thus, I think 1.47 Mbp or approx. 1.5 Mbp would be sufficient. (in P14, 15, 17, etc. I found similar description)

Reviewer #3:

Remarks to the Author:

The paper by Buessecker et al entitled An essential role for tungsten in the ecology and evolution of a novel lineage of anaerobic, thermophilic Archaea, deals with the enrichment and characterization of a novel Aigarchaeota lineage, *W. gerlachensis*, in respect of its W-dependency. The authors performed a myriad of experimental techniques to enrich and study the metabolic abilities of *W. gerlachensis* in terms of sugar and W-dependency. Based on the analysis of its genomic content and the results obtained, by analogy with existing literature, a model for the organism energy conservation is proposed. In addition, the authors collected samples from different locations and assembled/recruited/annotated 77 high-quality closely related genomes and performed ancestral character state reconstructions at different evolutionary depths, concluding that the W-dependency is an ancient trait of the lineage. Moreover, the authors propose a revised taxonomic classification of this family and briefly characterised the main physiological features of

the new lineages, based on selected genes of interest.

This is a long paper in which it is obvious the amount of work as well as synergies between both experimentalists and computational scientists. In this review I will focus mainly on the computational work and evolutionary considerations as well as in the overall organisation of the paper.

1- I think the paper should be restructured, since it is too long and many of the results are only briefly addressed in supplementary material, while in the main text, although having appreciated to read so many biological details, some analysis, are overly explained.

2- For instance, both phylogenetic analysis and structural modelling give some overlapping information: the classification of the different W-dependent Ferredoxin oxidoreductases according to its type.

3- On the contrary, even in supplementary information, the analysis of the ancestral content is discussed and based on selected KOs. How the selection of KO numbers was performed by the authors is unclear to me, especially since other genes are briefly discussed within the text in terms of synteny (not in terms of ancestral genomic content). Understanding that not all phylogenies for each single gene were computed, at least the predicted genomic content of the ancestral lineage should be discussed in main text. Moreover, even in supplementary information, this part is discussed in terms of comparative genomics and not in terms of ancestral reconstruction. In alternative, this part can be removed from the manuscript.

4- Another example relates with the mbh complex, first referred to in line 355, when the model proposed is explained, but only in line 400 we learn that one is found in the genome of *W. gerlachensis*, and a few lines after, that *P. furiosus*, has 2. At no given point we get information regarding to which of the two type 4 [NiFe] energy-conserving hydrogenase from *P. furiosus* is *W. gerlachensis* mbh-mrp more closely related. Since mbh and mbx/s use different substrates, it has consequences in terms of the model proposed.

5- In the Distribution of tungstoenzymes across the family Wolframiraptoraceae, either remove (or move to a dedicated section) the second paragraph related with the metabolic potential of other lineages as accessed by presence/absence of marker genes and/or ancestral state reconstructions.

Minor:

Last sentence from the introduction: please rephrase.

Line 355: „With the information gained from the protein phylogenetic and structural analyses, the physiological dependence on W was investigated. We propose a model...” = The model proposed is also based on the genomic content of the organism, and not only on the presence of the different W-dependent Oxidoreductases. Please rephrase

Methods: Protein structure modeling – it is stated that Oxidoreductase sequences were modeled . I think the authors mean W-dependent oxidoreductases.

Reviewer #4:

Remarks to the Author:

I was asked to comment on the NanoSIMS parts of this manuscript. The NanoSIMS methodology and interpretation is sound, carried out well, and in general presented clearly. I have a few minor

comments for the authors to consider.

Please clarify "probe-negative" and "probe-positive" in the caption of Fig. 3. This terminology is not used elsewhere in the manuscript and it would help the reader to understand what is meant.

The NanoSIMS and CARD-FISH images in Fig 3b are not explained in the text and as NanoSIMS is not a particularly well known technique it would help the reader to know how to interpret these images and what they are meant to show.

Did the authors perform ICP-MS on the cells themselves to determine W content?

Responses to Reviewers

An essential role for tungsten in the ecology and evolution of a novel lineage of anaerobic, thermophilic Archaea

Buessecker, Palmer et al., resubmitted to *Nature Communications* on March 3, 2022

Responses to Reviewer 1

Comments on Buessecker et al. “An essential role for tungsten in the ecology and evolution of a novel lineage of anaerobic, thermophilic Archaea”

I am not specialized in microbial cultivation techniques and have not worked with stable isotopes, nanoSIMS and CARD-FISH. I have experience with metabolic pathway analyses with aerobic bacteria but am not familiar with anaerobes and archaea. So I am not qualified to comment on these aspects. I do have rich experience with microbial evolutionary genomics, so my detailed suggestions are limited to these parts.

The evolutionary genomic analyses are not solid, nor complete, and the evolutionary story inferred from those analyses is not supported. There are a lot of discussion on the benefit of using W over Mo as a metal cofactor for microbes living in the early anoxic Earth before the GOE (2.33 Ga ago). These discussions are important and interesting. Without these, the part left is just the finding of W requirement by *Wolframiraptor gerlachensis*, and the potential (not fully consolidated) biochemical mechanism underlying this W requirement, which is important but represents an incremental advancement, certainly not sufficient to make a very attractive story. Note that obtaining the enrichment culture of *Wolframiraptor gerlachensis* is not new, as similar thing of a sister group has already been reported, which was noticed by the authors.

Thank you for your detailed review. We appreciate the reviewer’s concern regarding the evolutionary genomic analyses and the robustness of these analyses. In this particular study, we limited the evolutionary genomic analyses to phylogenomic analyses, at both the domain- and the family-level, and ancestral character state reconstructions (please see further responses below) coupled with phylogenetic inference for the genes of interest. To ensure the most reliable and robust phylogenies in both of these cases, and still ensure feasibility at the scale employed here, maximum-likelihood analyses were used for all phylogenetic analyses, coupled with two independent branch support estimates as employed by IQ-Tree or with traditional bootstrapping implemented in RAxML from 1,000 pseudoreplicates in each case. We thus endeavored to reach a balance between 1) as comprehensive as feasible taxon selection in phylogenetic analyses, 2) obtaining the best possible multiple sequence alignments, and 3) applying the appropriate evolutionary model for each molecular marker analyzed. In order to address point 1), we included all publicly available genome sequences in the Genome Taxonomy Database (GTDB) R95, for the *Caldarchaeales* (syn. *Aigarchaeota*), as well as appropriate outgroup taxa in the *Nitrososphaeria* for the family-level phylogenies (resulting in a taxon set of 98 taxa, **Fig. 5**), and representatives for all archaeal phyla (as determined in Dombrowski et al., 2020, *Nature Communications*), together with the newly determined genomes for the domain-level phylogeny (resulting in a taxon set of 272 genomes, **Fig. S10**). Additionally, for single gene trees, all homologs were identified from the non-redundant database, and were included in the analysis, along with reference sequences from UniProt. This resulted in varying taxon sets per marker but ranged between approximately 100 and 2,650 taxa per analysis. As such, we

included the most comprehensive taxon sets possible for all phylogenies. To address point 2), we aligned all sequences analyzed, with the iterative alignment algorithms employed in MAFFT, and where appropriate, with the DASH algorithm that performs alignments based on structurally homologous protein domains. To address point 3) the best-fit evolutionary model of amino acid substitution for each molecular marker was determined and applied during phylogenetic inference, including in the cases where concatenation of sequences were performed, where the concatenated matrices were partitioned per marker sequence and the best-fit model applied to each partition. Overall, as an absolute tungsten requirement is rarely observed, and members belonging to the archaeal order *Caldarchaeales* have reportedly only once been cultivated in stable co-cultures and not characterized before, we believe the findings reported here are still substantial and significant, particularly in light of large research efforts afforded to cultivation attempts of novel lineages.

The deep time evolution story that the authors tried to present and highlight builds on two key assumptions: (1) the last common ancestor (LCA) of Wolframiraptoraceae used W, and (2) the LCA evolved before GOE. There are a number of important problems. Apparently, testing the assumption (2) requires estimating the age of Wolframiraptoraceae (the crown group of Wolframiraptoraceae), which is completely missing. This is possible, though few fossils are available in this lineage or even in Archaea. For example, the authors may consider building a tree that include archaea and eukaryotes, and use the rich fossils contained in eukaryotic lineages to help estimate the age of Wolframiraptoraceae. This of course requires additional expertise, but again this analysis is important and feasible. For testing the assumption (1), the authors employed COUNT. This is far from being sufficient. There have been a lot of concerns regarding using birth-and-death models, based on which COUNT was developed, to infer the gene gain and loss events along an evolutionary tree. It's essential to try a completely different type of methods, in particular those based on gene tree species tree reconciliation. The birth-and-death models and the tree topological incongruence based methods often give different results. The authors can be confident on the inferred events supported by both types of methods. For any inconsistent predictions, additional analyses must be followed to convince people if one prediction is favored over the other by the authors.

We appreciate this useful input. However, the first assumption as stated by the reviewer is partially correct and there is no evidence that the LCA of the family *Wolframiraptoraceae* evolved before the GOE. In fact, as a family-level lineage, if the basis of relative evolutionary divergence is correct (i.e., that all taxa representing similar taxonomic ranks co-existed in deep-time), then it is likely that this family could have evolved later in the Proterozoic, at the broadest estimates between 400 Mya and 1,400 Mya, akin to other families within the phylum *Thermoproteota* (Marin et al., 2016, *Molecular Biology and Evolution*; Colman et al., 2018, *The ISME Journal*; Yang et al., 2021, *Molecular Biology and Evolution*). For example, i) the largest efforts at dating prokaryotic evolutionary events puts the divergence of other families in the phylum at ~850-1,400 Mya, specifically the *Desulfurococcaceae* (1,385 Mya), *Sulfolobaceae* (852 Mya), and *Thermoproteaceae* (908 Mya) (Marin et al., 2016, timetree.org); ii) the LCA of order-level lineages of closely related classes (*Sulfolobales* and *Thermoplasmatales*) were estimated as evolving between ~800-1,100 Mya (Colman et al., 2018); and iii) the emergence of the *Nitrosopumilaceae*, also known as the marine ammonia-oxidizing archaea (AOA) and members of the same class as the *Caldarchaeales*, was most recently estimated as diverging from terrestrial AOA at approximately 509 (~412 to 629) Mya (Yang et al., 2021). Thus, although some variation in date estimates are observed between these studies, none of these estimates would be consistent with family-level lineages evolving before the GOE. As such, dating analyses were not included in this study as we believe insufficient taxonomic representation within the order *Caldarchaeales* (as stated in the

manuscript, only three high-quality genomes for the rest of the order *Caldarchaeales*, outside of the *Wolframiraptoraceae* are included in the GTDB R95 dataset) and the remainder of the class *Nitrososphaeria*, is available at present to improve on existing estimates based on closely related lineages. Thus, in our opinion, robust molecular dating analyses is not yet feasible for this lineage with the current limited genomic representation, and these analyses would be better suited once increased genomic diversity is discovered and available to refine existing estimates. Additionally, as we do not hypothesize that the LCA evolved before the GOE, we do not believe these analyses would contribute greatly to this manuscript.

We appreciate the reviewer's input on the use of COUNT and agree that birth-and-death models alone are not sufficient for inferring evolutionary events in this lineage. For this reason, robust phylogenetic analyses for all individual molecular markers were also included in the manuscript and the ancestral reconstruction results were interpreted at the hand of these phylogenies. However, we thank the reviewer for the suggestion of including gene tree/species tree reconciliation analyses and have included these in the revised manuscript (see Fig. 5, L427-446). As interpretations of the potential evolutionary scenarios presented in the Supplemental Text were done at the hand of the robust phylogenies for those markers analyzed, the description of these evolutionary scenarios in the supplemental results did not change markedly, although we did expand where applicable. However, we realize that the view presented by the birth-and-death model ancestral reconstruction of character states figure in the main text may have provided an unintentionally simplified view of the evolution of the markers, and as such, we have replaced the COUNT-based ancestral character state reconstruction figure for the tungsten-associated markers in the main text with a summary of the gene tree/species tree reconciliation analyses to reflect the complex evolutionary history of these markers (revised **Fig. 5**).

Another problem of this manuscript is its flow. One reason is that it contains many taxonomic names. It's difficult for microbiologists who don't work on these groups to follow the paper. It's even more difficult for people working in other areas.

We appreciate that several novel names are being proposed and are introduced in this manuscript, and that this work represents a complex and large body of work. However, we believe that referring to these newly proposed taxa by their new names, in fact, assists readers in following the discussion better than using existing and/or additional novel alphanumeric strings as currently used in the databases. However, we have tried to address this concern by moving specific discussion regarding the different genera in the family to the supplement (pages 18 and 19), and improved the flow and conciseness of the manuscript (word count of the main text dropped by almost 20% to 5,566 words).

Responses to Reviewer 2

The authors successfully revealed the carbon and energy metabolisms of a previously uncultivated lineage of *Caldarchaeales*, and the impacts of tungsten on them. Moreover, the authors discussed about the importance of tungsten on the early life on Earth, and the historical importance of the metabolisms related to tungsten in thermophilic life inhabiting geothermal and hydrothermal ecosystems. The carbon and energy metabolisms of the archaeon was unveiled by holistic approach applicable for uncultivated microorganisms. The experimental design and manuscript are generally well organized. However, very important previous reports presenting the significance of tungsten on carbon and energy metabolisms in hyperthermophilic archaeon *Pyrobaculum* are absent (e.g. Hagedoorn et al. 2005; de Vries

et al. 2010, Braseb et al. 2014 and Habib and Hoffman 2017). Pyrobaculum is one of the most predominant hyperthermophilic archaeon in high temperature geothermal and coastal hydrothermal ecosystems with neutral or slightly acidic pH. Thus, the discussion related to evolution and early life in this study might be influenced by these studies.

(Note that Pyrobaculum has been recognized as a member of Crenarchaeota that is a sister clade of Aigarchaeota. In the novel taxonomic scheme (Rinke et al. 2021) used in this study, Crenarchaeota is classified as Thermoproteia.)

In addition, I would suggest to compare the carbon and energy metabolisms in other (hyper-)thermophilic organisms such as Thermococcales and Thermotoga. The linkage between carbohydrate degradation and ATP generation systems of these organisms are similar to Wolframiraptor. Thus, such comparison would be beneficial in the discussion of the evolution of early thermophilic life and metabolisms related to tungsten.

We are very grateful for the additional sources the reviewer is suggesting which together with the existing body of literature cited provide an extensive overview of related work done previously. We included citations of studies on Pyrobaculum (Hagedoorn et al. 2005; de Vries et al. 2010 and Brasen et al. 2014 in L73, and Habib and Hoffman 2017 in L133). To keep the text concise, we added a short insight into the metabolic comparison to the marine thermophile *Thermotoga maritima* (L355). “[...] The carbon and energy metabolism in that bacterium are similar to that of *W. gerlachensis*: Formation of pyruvate via the Embden-Meyerhof pathway, production of organic acids and H₂ and therefore coupling of W enzymatic activity to a hydrogenase and alternative reduction of S⁰ to H₂S by H₂. [...]” This discussion is indeed beneficial placing the W-dependent anaerobic heterotrophic metabolism of *W. gerlachensis* into broader context and we thank the reviewer for this idea.

L63: “fermentation” has versatile meanings, and is confusing for readers considering the foci of the manuscript.

We replaced “fermentation” with “anaerobic carbohydrate degradation”.

L76: See the major comments.

We included Hagedoorn et al. 2005; de Vries et al. 2010 and Brasen et al. 2014 in L73, and Habib and Hoffman 2017 in L133.

L78: Euryarchaeota is not applicable in the taxonomy used in this study.

Thank you for pointing this out. We have replaced the term *Euryarchaeota* with the *Methanobacteriota*, the phylum name associated with the *Thermococcales* in the current GTDB.

Fig 1: Please briefly present the methods used to obtain absolute abundance of both total microbial communities and *W. gerlachensis* more clearly. In M&M and supplementary material, I could not identify the method to obtain total microbial abundance.

We included the method information for each data set shown in Fig. 1a and 1b in the caption to Fig. 1.

L159: hyperthermophile

We corrected this.

L195: “difference of the systems that require W” would also explain.

We added this in addition to the other two possible explanations.

Fig.2: I think normalization is necessary to compare ASV abundance but not mentioned.

Thank you for addressing this. We normalized all sample raw ASV reads to the minimum number of 5739 reads using SRS (Beule L, Karlovsky P. 2020. Improved normalization of species count data in ecology by scaling with ranked subsampling (SRS): application to microbial communities. PeerJ 8:e9593 <https://doi.org/10.7717/peerj.9593>). We used the SRS method over rarefying because of known problems with rarefying microbial sequence data (McMurdie PJ, Holmes S (2014) Waste Not, Want Not: Why Rarefying Microbiome Data Is Inadmissible. PLoS Comput Biol 10(4): e1003531, <https://doi.org/10.1371/journal.pcbi.1003531>). Because the emerging trends were identical to the raw read data, we kept the raw reads for Figure 2 and refer to SRS-normalized data in Supplementary File S5.

L621- and 631-: Information about primer sets is missing.

We included the references to the information in the supplements (Table S4).

L625: Did the authors confirmed the amplicon by gel electrophoresis?

We didn't do this for every qPCR performed. However, in initial qPCR runs, we did run some products out on a gel to confirm the size, and had them sequenced by Sanger sequencing to confirm that the product was from Aigarchaeota Group 4 (Wolframiraptor).

L731: Delete Ca.

We deleted it.

L767: Please complete the data deposition prior to the revision.

This has been done, and all genomes are publicly available in several data repositories under the accession numbers indicated in Table S6.

Supplementary Text

P12: What does “Sulfide-dependent respiration” mean?

This has been adjusted to specifically refer to sulfide (Sqr) or sulfur (DsrAB) oxidation, or heterodisulfide (HDR) or sulfite (DsrAB) reduction where relevant.

P13: Pacific Ocean

Thank you, corrected.

It is not necessary to present accurate sequence length of incomplete binned genome because the lengths must be influenced by parameters in assembly and binning. Thus, I think 1.47 Mbp or approx. 1.5 Mbp would be sufficient. (in P14, 15, 17, etc. I found similar description)

Although we share this sentiment with the reviewer, it is common to report the full length of genomes (even incomplete MAGs), especially since this also provides a link specifically to which assembly of the data serves as the type for the taxon. Thus, although we are in agreement that the full genome size of an individual within the population cannot reasonably be determined, the full reported length for these assemblies were kept in the supplemental text.

Responses to Reviewer 3

The paper by Buessecker et al entitled An essential role for tungsten in the ecology and evolution of a novel lineage of anaerobic, thermophilic Archaea, deals with the enrichment and characterization of a novel Aigararchaeota lineage, *W. gerlachensis*, in respect of its W-dependency. The authors performed a myriad of experimental techniques to enrich and study the metabolic abilities of *W. gerlachensis* in terms of sugar and W-dependency. Based on the analysis of its genomic content and the results obtained, by analogy with existing literature, a model for the organism energy conservation is proposed. In addition, the authors collected samples from different locations and assembled/recruited/annotated 77 high-quality closely related genomes and performed ancestral character state reconstructions at different evolutionary depths, concluding that the W-dependency is an ancient trait of the lineage. Moreover, the authors propose a revised taxonomic classification of this family and briefly characterised the main physiological features of the new lineages, based on selected genes of interest.

This is a long paper in which it is obvious the amount of work as well as synergies between both experimentalists and computational scientists. In this review I will focus mainly on the computational work and evolutionary considerations as well as in the overall organisation of the paper.

1- I think the paper should be restructured, since it is too long and many of the results are only briefly addressed in supplementary material, while in the main text, although having appreciated to read so many biological details, some analysis, are overly explained.

We made great effort to improve clarity and conciseness of the paper. In the section “A stable enrichment culture of a novel, anaerobic member of the *Caldarchaeales*” we reduced text that included too much detail of the culture description (especially on alternative metabolites tested). We cut this section by approximately 150 words. In the section “W is essential to maintain growth of *W. gerlachensis*” we extracted redundant information (mostly on enrichment descriptions and W requirements) and shortened the passages considering different potential limiting factors for *W. gerlachensis* growth. Here, we get faster to the main message of this paragraph, that is, that the MAG suggested W transporters, and the only trace element present in the spring water (and not elsewhere in the substrate/media) was W. This adjustment bridges the culture description more directly to the follow-up experiments. We are now also more brief in describing community effects of tungsten limitation (Fig. 2). In the section “Xylose is the preferred carbohydrate monomer for *W. gerlachensis*” we focused on the FISH-NanoSIMS results relevant to *W. gerlachensis* and left out text passages about observations not well supported by replication (reducing text by about 140 words). We rewrote the legend to Fig. 3 for better clarity and to include more context to the method used (as Reviewer 4 suggested as well). In the following section “W-dependent ferredoxin oxidoreductases putatively drive aldehyde metabolism” we took out information about the anomalous GAPOR (P16) protein structure that might have been interesting but unnecessary for the main

story. In the section “Distribution of tungstoenzymes across the family *Wolframiraptoraceae*” we completely deleted the paragraph on the genomic content of MAGs from taxa other than *W. gerlachensis*. In order to retain the information for more specialized readers, we moved the paragraph into the supplements, which reduced the section by approximately 170 words. We rewrote the last paragraph before the Discussion section, also owing to new conclusions drawn from the additional gene tree/species tree reconciliation analyses. To link this last component of our study better to the main objective, we restructured also the key (first) sentence of the paragraph (now in L427). Generally, we also avoided use of unnecessary “fill” words, chose shorter alternatives for expressions and limited the use of novel taxonomic names to one section. Overall, the word count of the main text dropped by almost 20% to 5,566 words, including figure legends.

2- For instance, both phylogenetic analysis and structural modelling give some overlapping information: the classification of the different W-dependent Ferredoxin oxidoreductases according to it's type.

Although we can appreciate the fact that both approaches provided the overall same conclusions regarding functions for the W-dependent ferredoxin oxidoreductases, we believe that coming to these conclusions with different methodologies provides more support for hypotheses regarding these very novel enzymes. Additionally, Reviewer 1 suggested on a different part of the study that “It's essential to try a completely different type of methods,...” so that “The authors can be confident on the inferred events supported by both types of methods.”. Thus, although this can be described as “overlapping information”, we rather consider this data as complementary information. This approach provides consistent results from two different analyses to infer the identity/function of W-dependent oxidoreductase sequences, and thus increases the robustness in the functional assignment of these enzymes. As the results from both the phylogenetic analysis and structural modelling support each other and lead to the same conclusions, we are of the opinion that including both in the main text does not disorganize the paper or make it harder to follow. In fact, as the identity of the W-dependent oxidoreductases is an important aspect of the paper that is well supported through different lines of evidence, this provides a robust foundation for the subsequent analyses on these genes (distribution within the family, ancestral reconstruction, and the newly added gene tree/species tree reconciliation analyses). Thus, we believe that the space afforded for both these analyses in the manuscript is well used considering the value gained from their complementarity.

3- On the contrary, even in supplementary information, the analysis of the ancestral content is discussed and based on selected KOs. How the selection of KO numbers was performed by the authors is unclear to me, especially since other genes are briefly discussed within the text in terms of synteny (not in terms of ancestral genomic content). Understanding that not all phylogenies for each single gene were computed, at least the predicted genomic content of the ancestral lineage should be discussed in main text. Moreover, even in supplementary information, this part is discussed in terms of comparative genomics and not in terms of ancestral reconstruction. In alternative, this part can be removed from the manuscript.

Although the full ancestral character state reconstruction was performed on the entire genomes of members of the *Wolframiraptoraceae*, we focused specifically on genes associated with energy metabolism and W-associated enzymes as this fit well with the rest of the manuscript foci. Additionally, due to the current length of the manuscript, as this reviewer points out, there was limited space to expand on the overall ancestral content of the genomes in the main text. Furthermore, we believe that future studies among the different families of the *Caldarchaeales* would be more valuable and provide a more robust comparison to discuss ancestral genome content more fully, and this is intended for a future study.

We do however agree that the ancestral gene content for the energy metabolism genes analyzed here are insufficiently described in the supplementary text, and as such, have expanded on these aspects in the paragraphs dealing with this specifically. Thank you for this suggestion.

4- Another example relates with the mbh complex, first referred to in line 355, when the model proposed is explained, but only in line 400 we learn that one is found in the genome of *W. gerlachensis*, and a few lines after, that *P. furiosus*, has 2. At no given point we get information regarding to which of the two type 4 [NiFe] energy-conserving hydrogenase from *P. furiosus* is *W. gerlachensis* mbh-mrp more closely related. Since mbh and mbx/s use different substrates, it has consequences in terms of the model proposed.

We now specify the hydrogenase in *W. gerlachensis* as membrane-bound and proton-translocating in L347. In L303 we also explicitly added that Mbh is contained in the *W. gerlachensis* genome to make the relevance of the focus on this gene complex more clear earlier in the section.

5- In the Distribution of tungstoenzymes across the family Wolframiiiraptoraceae, either remove (or move to a dedicated section) the second paragraph related with the metabolic potential of other lineages as accessed by presence/absence of marker genes and/or ancestral state reconstructions.

In staying with the suggestions to improve conciseness and decrease the length of the manuscript, we have removed this paragraph from the manuscript and included these details in the additional supplementary text.

Minor:

Last sentence from the introduction: please rephrase.

We adjusted this sentence (L74): “Our investigation is unique in showing long-term W-dependent growth dynamics in a mixed culture, identifying W-requiring archaea outside of the *Methanobacteriota*, and demonstrating large-scale expansions of W-dependent enzymes in a microbial lineage.”

Line 355: „With the information gained from the protein phylogenetic and structural analyses, the physiological dependence on W was investigated. We propose a model...” = The model proposed is also based on the genomic content of the organism, and not only on the presence of the different W-dependent Oxidoreductases. Please rephrase

We added here that the genomic content was also used as a source of information as suggested.

Methods: Protein structure modeling – it is stated that Oxidoreductase sequences were modeled . I think the authors mean W-dependent oxidoreductases.

We adjusted this accordingly.

Responses to Reviewer 4

I was asked to comment on the NanoSIMS parts of this manuscript. The NanoSIMS methodology and

interpretation is sound, carried out well, and in general presented clearly. I have a few minor comments for the authors to consider.

Please clarify "probe-negative" and "probe-positive" in the caption of Fig. 3. This terminology is not used elsewhere in the manuscript and it would help the reader to understand what is meant.

We changed the expression omitting "probe-negative" and "probe-positive".

The NanoSIMS and CARD-FISH images in Fig 3b are not explained in the text and as NanoSIMS is not a particularly well known technique it would help the reader to know how to interpret these images and what they are meant to show.

We now added a more explanatory sentence in L262: "The higher relative enrichment of ^{13}C and ^{15}N in *W. gerlachensis* corresponds to higher single-cell anabolic activity compared to other surrounding cells."

Did the authors perform ICP-MS on the cells themselves to determine W content?

We thank the reviewer for this consideration and we agree, it would be valuable to assess the cellular W content. However, measuring W content of the bulk cells in the enrichment might not reveal much because *W. gerlachensis* is just a fraction of the population and other cells may or may not take up W. Besides using ICP-MS, we attempted to quantify the intracellular incorporation of W by *W. gerlachensis* cells using NanoSIMS, but the relatively low ionization yield of W, using both cesium and oxygen ion sources, combined with an apparent low cellular W content (in *W. gerlachensis* as well as in *Pyrococcus furiosus* controls), led to inconclusive results.

Reviewers' Comments:

Reviewer #1:

Remarks to the Author:

The revised manuscript is clearer and improved. Some of my comments have been addressed, such as using gene tree - species tree reconciliation, but other important parts remain unaddressed. In particular, the authors put their findings in context of GOE, but they are not willing to perform molecular dating analysis to make a stronger connection between the evolution of this archaea lineage and the GOE, the most important time-calibrated Earth event. While this is certainly disappointing, the paper in its present form is still acceptable for publication as it has other important advances.

Reviewer #2:

Remarks to the Author:

The authors have revised satisfactory. I would provide minor comments as shown below.

L48: "hydrothermal and geothermal ecosystems" instead of "marine and terrestrial thermal waters"

L73: "W is important" remind reader that W is required for growth. The structure of the sentence is confusing. I think "including methanogens and representatives of Thermococcales and Thermoproteales" would be better.

L75: Add "Methanobacteriota (formerly Euryarcgaeota)"

L77: 2nd W-requiring archaea outside of

Fig 1. b. and Fig 2 b and c: ml⁻¹ medium

L133: *Pyrococcus furiosus* in Thermococcales.

L248: suggesting instead of indicating

L286: *Caldicellulosiruptor* in Clostridia

L358: marine hyperthermophilic bacteria

L475: "anaerobic" instead of "fermentative"

Fig S8: Lower taxonomic identification instead of Thermoproteia is helpful for readers if possible.

Supplementary text

P6: Thermoproteia (formerly Crenarchaeota)

P8 and through the manuscript: Bacillota (formerly Firmicutes) instead of Firmicutes.

P8-9: HGT between Bacteria and Archaea should be discussed more clearly, or bacterial lineages should be defined in supplementary figures. It is difficult to distinguish between Archaea and Bacteria for most of the readers in this journal.

P11: Use italic for cytochrome "bd".

P12: How about nitrate reductase? W-dependent nitrate reductase has been reported in *Pyrobaculum*. If the authors find it in this lineage, it is better to be mentioned.

Reviewer #3:

Remarks to the Author:

The revised version of the manuscript addressed all of the comments of the reviewers and it is highly improved. Namely, by performing tree reconciliation methods as suggested by Reviewer 1, the acquisitions of the different proteins could be further clarified.

I have no further comments on the paper.

Responses to Reviewers, 2nd round

An essential role for tungsten in the ecology and evolution of a novel lineage of anaerobic, thermophilic Archaea

Buessecker, Palmer et al., resubmitted to *Nature Communications* on May 27, 2022

Responses to Reviewer 1

The revised manuscript is clearer and improved. Some of my comments have been addressed, such as using gene tree - species tree reconciliation, but other important parts remain unaddressed. In particular, the authors put their findings in context of GOE, but they are not willing to perform molecular dating analysis to make a stronger connection between the evolution of this archaea lineage and the GOE, the most important time-calibrated Earth event. While this is certainly disappointing, the paper in its present form is still acceptable for publication as it has other important advances.

We thank the reviewer for their time and appreciate their suggestions which improved the manuscript substantially.

Responses to Reviewer 2

The authors have revised satisfactory. I would provide minor comments as shown below.

We thank the reviewer for their time and appreciate their detailed feedback which improved the manuscript substantially.

L48: “hydrothermal and geothermal ecosystems” instead of “marine and terrestrial thermal waters”

Done.

L73: “W is important” remind reader that W is required for growth. The structure of the sentence is confusing. I think “including methanogens and representatives of Thermococcales and Thermoproteales” would be better.

We included this suggestion.

L75: Add “Methanobacteriota (formerly Euryarcgaeota)”

Done.

L77: 2nd W-requiring archaea outside of

We do not claim to be the first as in previous versions and we would prefer the current sentence.

Fig 1. b. and Fig 2 b and c: ml -1 medium

We would prefer the current form of the units. That the volume refers to ml culture can be inferred from the caption.

L133: *Pyrococcus furiosus* in Thermococcales.

We added it.

L248: suggesting instead of indicating

Done.

L286: *Caldicellulosiruptor* in Clostridia

We added it.

L358: marine hyperthermophilic bacteria

We would prefer the current formulation.

L475: “anaerobic” instead of “fermentative”

This suggestion is in conflict with a previous reviewer suggestion and so we would prefer to leave the current word choice.

Fig S8: Lower taxonomic identification instead of Thermoproteia is helpful for readers if possible.

Done.

Supplementary text

P6: Thermoproteia (formerly Crenarchaeota)

We added it.

P8 and through the manuscript: Bacillota (formerly Firmicutes) instead of Firmicutes.

Done.

P8-9: HGT between Bacteria and Archaea should be discussed more clearly, or bacterial lineages should be defined in supplementary figures. It is difficult to distinguish between Archaea and Bacteria for most of the readers in this journal.

We now discuss this starting on page 29 of the Supplementary Information.

P11: Use italic for cytochrome “bd”.

Done.

P12: How about nitrate reductase? W-dependent nitrate reductase has been reported in *Pyrobaculum*. If the authors find it in this lineage, it is better to be mentioned.

For the sake of conciseness (the supplementary information comprises 45 pages), we would prefer to leave it at that extent. We do thank the reviewer for their excellent ideas nevertheless.

Responses to Reviewer 3

The revised version of the manuscript addressed all of the comments of the reviewers and it is highly improved. Namely, by performing tree reconciliation methods as suggested by Reviewer 1, the acquisitions of the different proteins could be further clarified.

I have no further comments on the paper.

We thank the reviewer for their time and appreciate their feedback which improved the manuscript substantially.